# Quinones as Promising Compounds against Respiratory Viruses: A Review

**DOI:** 10.3390/molecules28041981

**Published:** 2023-02-20

**Authors:** Ivan Chan-Zapata, Rocío Borges-Argáez, Guadalupe Ayora-Talavera

**Affiliations:** 1Unidad de Biotecnología, Centro de Investigación Científica de Yucatán, Chuburná de Hidalgo, Merida 97205, Mexico; 2Departamento de Virología, Centro de Investigaciones Regionales “Dr. Hideyo Noguchi”, Universidad Autónoma de Yucatán, Paseo de Las Fuentes, Merida 97225, Mexico

**Keywords:** influenza, SARS-CoV-2, naphthoquinone, anthraquinone, benzoquinone, antivirals, natural compounds, synthetic quinones

## Abstract

Respiratory viruses represent a world public health problem, giving rise to annual seasonal epidemics and several pandemics caused by some of these viruses, including the COVID-19 pandemic caused by the novel SARS-CoV-2, which continues to date. Some antiviral drugs have been licensed for the treatment of influenza, but they cause side effects and lead to resistant viral strains. Likewise, aerosolized ribavirin is the only drug approved for the therapy of infections by the respiratory syncytial virus, but it possesses various limitations. On the other hand, no specific drugs are licensed to treat other viral respiratory diseases. In this sense, natural products and their derivatives have appeared as promising alternatives in searching for new compounds with antiviral activity. Besides their chemical properties, quinones have demonstrated interesting biological activities, including activity against respiratory viruses. This review summarizes the activity against respiratory viruses and their molecular targets by the different types of quinones (both natural and synthetic). Thus, the present work offers a general overview of the importance of quinones as an option for the future pharmacological treatment of viral respiratory infections, subject to additional studies that support their effectiveness and safety.

## 1. Introduction

Respiratory viruses are a group of pathogens that elicit upper or lower respiratory tract infections [1]. Upper respiratory tract infections affect the nasal cavity through the larynx and have common symptoms, such as cough, sore throat, nasal congestion, sneezing, rhinorrhea, sinus pain, myalgia, headache, fever, chills, and loss of appetite. Lower respiratory tract infections occur below the larynx and can also cause bronchitis, bronchiolitis, and pneumonia [2]. Human respiratory viruses cause these symptoms after infecting and replicating in cells of the respiratory tract. Subsequently, these pathogens are transmitted mainly through the respiratory secretions of infected people by direct (physical) or indirect contact (contaminated objects or surfaces), as well as by droplets or aerosols [3,4].

Viral respiratory diseases represent an important problem for global public health with an enormous economic burden. These respiratory infections are among the most frequent illness in humans, mainly in children, older adults, and immunosuppressed people, making them the most vulnerable populations. In children and older adults, viruses cause 95% and 40% of all respiratory infections, respectively. In addition, respiratory tract infections are associated with significant mortality worldwide. According to the World Health Organization (WHO), in 2019, lower respiratory infections ranked fourth among the leading causes of death in the world, with 2.6 million deaths. WHO also ranked respiratory infections as the second leading cause of death worldwide in children <5 years of age, estimating 1.9 million deaths annually for complications related to acute respiratory infections [5,6,7]. Respiratory viruses include the parainfluenza viruses (PIVs), human metapneumoviruses (HMPVs), respiratory syncytial viruses (RSVs), adenoviruses (AdVs), rhinoviruses (RVs), bocaviruses (BoVs), influenza viruses, and coronaviruses (CoVs) [8]. The aim of this work was to describe the activity of quinones against respiratory viruses such as influenza viruses, CoVs, PIV, RSV, and HRV. Likewise, we review and summarize the properties of these compounds over the viral targets.

## 2. Respiratory Viruses

### 2.1. Parainfluenza Virus (PIV), Human Metapneumovirus (HMPV), and Respiratory Syncytial Virus (RSV)

PIV, HMPV, and RSV are respiratory viruses belonging to the *Paramyxoviridae* family. These pathogens are enveloped viruses containing a non-segmented negative-sense single-stranded RNA (ssRNA) genome. Globally, these viral agents are characterized by causing significant morbidity and mortality, mainly among children in developing countries [9,10]. RSV is among the most important pathogens causing lower respiratory tract infections in childhood. However, immunosuppressed people and older adults are also at high risk of developing complications from RSV infections. There is a single RSV serotype with two antigenic subtypes (A and B), which circulate together, but only one subtype predominates. Although RSV mortality is more common in developing countries, the social and economic burden associated with this virus is high worldwide [11,12].

Human PIVs (HPIVs) include four serotypes (HPIV1-4) and represent one of the main causes of acute respiratory infections. These viruses affect people of all ages but cause lower respiratory tract infections that can lead to serious illness in infants and young children. HPIVs infection rates are highest in children <5 years of age, followed by patients >60 years of age. In healthy young adults, this disease is usually mild and restricted to the upper respiratory tract [13,14].

HMPVs include two genotypes (A and B) divided into five lineages (A1, A2a, A2b, B1, and B2). Similar to previous viruses, HMPVs cause respiratory infections of varying severity and circulate around the world throughout the year, but the epidemics caused by these pathogens predominate in spring and winter. From 1995 to 2016, 42,464 cases of infections were reported in countries such as Canada, the United States, Mexico, Spain, Austria, Scotland, South Africa, South Korea, and Taiwan [15,16,17].

### 2.2. Adenovirus (AdV)

AdVs are non-enveloped icosahedral pathogens that possess a double-stranded DNA (dsDNA) genome and belong to the *Adenoviridae* family [18]. These viruses are divided into seven species (A to G) with 103 recognized types. Respiratory infections associated with human AdVs (HAdVs) are mainly caused by genera B (types 3, 7, 14, 21, and 55), C (types 1, 2, and 5), and E (types 4 and 41). Although most respiratory infections caused by these viruses are mild to moderate in severity, HAdV-B types 3, 7, 14, and 55 can cause severe infections and life-threatening outbreaks. In this sense, regional outbreaks of severe and fatal acute infections have been reported in Europe, Asia, and North America caused by HAdV-B7 or HAdV-B14 [17,19,20].

### 2.3. Rhinovirus (RV)

Human RVs (HRVs) belong to the *Picornaviridae* family and are non-enveloped viruses with positive-sense ssRNA. HRVs are classified into A, B, and C species, which have approximately 80, 32, and 57 types, respectively. RVs are among the most frequent respiratory viruses in humans, causing significant morbidity and high annual economic losses. These pathogens are responsible for the common cold and cause more than 50% of upper respiratory tract infections. HRVs are involved in lower respiratory illnesses such as pneumonia, bronchitis, and bronchiolitis, as well as exacerbations of chronic obstructive pulmonary disease (COPD) and asthma. These viruses circulate throughout the year and affect the entire world, although the highest peak of infections is observed in the autumn and declines in the summer, with HRV-A and HRV-C as the predominant species [21,22,23].

### 2.4. Bocavirus (BoV)

Recently, human BoVs (HBoVs) have appeared as new respiratory pathogens. These non-enveloped viruses are found in the *Parvoviridae* family and contain a linear negative- or positive-sense ssDNA genome. There are four types of HBoVs (HBoV1-4), and infections caused by these viral agents are more common during winter and spring, although they are present all year. HBoV infections are also present in people of all ages but are more frequent in children. Finally, although these viruses do not have a high global prevalence, fatal cases have been reported due to HBoV infections [24,25,26].

### 2.5. Influenza Virus

Influenza viruses are another group of enveloped respiratory viruses that belong to the *Orthomyxoviridae* family and contain a genome consisting of negative-sense ssRNA segments. The *Orthomyxoviridae* family includes four types of influenza viruses (A to D). Influenza A and B viruses mainly affect humans, being responsible for annual seasonal epidemics. Influenza B infections are highly contagious and sometimes lead to severe illness. In contrast, influenza A infections are the most common and cause mild to severe respiratory illness. Influenza C viruses are the least common and produce milder infections compared to influenza A and B, while there are no reports of influenza D viruses infecting humans [27,28].

Influenza B viruses are divided into two lineages: B/Yamagata/16/88 and B/Victoria/2/87 (commonly known as B/Yamagata and B/Victoria, respectively). Influenza A viruses are classified considering the hemagglutinin (HA) and neuraminidase (NA) subtypes located on their surface. There are 18 HA subtypes (H1-18) and 11 NA subtypes (N1-11), but three HA and two NA subtypes have circulated in humans, appearing in the population as the pandemic strains H1N1, H2N2, and H3N2 [29,30,31]. In the past century, the 1918 “Spanish” influenza pandemic was caused by an H1N1 strain (apparently derived from an avian source), the 1957 “Asian” influenza pandemic H2N2, and the 1968 “Hong Kong” influenza pandemic H3N2 (both emerging from reassortments between human and avian viruses) were responsible for 50 million, two million, and one million deaths around the world, respectively [32,33].

Later, in 2009, a new influenza A(H1N1) virus with gene segments from strains of avian, porcine, and human viruses emerged. The respiratory diseases of this influenza A(H1N1)pdm09 strain were responsible for around 201,200 deaths and affected more than 214 countries, causing the first influenza pandemic of the 21st century [34,35,36]. On the other hand, avian influenza has emerged as a threat to poultry, causing great economic losses. In addition, there are highly pathogenic avian influenza viruses with the ability to infect other mammals, such as pigs and humans, posing a threat to global public health due to the risk of a potential pandemic. Several outbreaks of avian influenza in humans have recently been reported, which have been attributed to close contact with infected birds or to a contaminated environment, such as markets where live birds are sold [37,38]. Worldwide, since 2003, the WHO has estimated 868 cases of human infections with avian influenza A(H5N1) virus in 21 countries, of which 457 resulted in fatal cases, while 1568 cases of human infections and 616 deaths associated with the avian influenza A(H7N9) virus have been reported since 2013 [39].

Influenza B viruses have mutations and limited animal reservoirs, giving them relative stability. Nevertheless, the possibility of genetic modification and increased pathogenicity is always latent, with the consequent generation of a pandemic strain of influenza B [40]. Currently, seasonal influenza epidemics are caused by influenza A(H1N1), A(H3N2), and B viruses. Globally, these seasonal epidemics result in about 3–5 million cases of severe illness and 290,000–650,000 deaths [41].

### 2.6. Coronavirus (CoV)

CoVs are enveloped viruses with a non-segmented positive-sense ssRNA. These viruses belong to the *Coronaviridae* family and are divided into four genera (alpha, beta, gamma, and delta CoVs). Gamma and delta CoVs cause avian CoV infections (birds are the natural reservoirs of these viruses), whereas mammalian CoV diseases are mainly associated with alpha and beta CoVs (bats and rodents are the natural reservoirs). There are seven human CoVs (HCoVs) that cause infections, four of which (HCoV-229E, HCoV-OC43, HCoV-NL63, and HCoV-HKU1) cause mild seasonal respiratory tract diseases around the world. In contrast, three highly pathogenic HCoVs have been identified this century: severe acute respiratory syndrome CoV (SARS-CoV), Middle East respiratory syndrome CoV (MERS-CoV), and severe acute respiratory syndrome CoV 2 (SARS-CoV-2) [42,43,44].

In November 2002, SARS-CoV caused the first coronavirus epidemic. This virus originating from China, caused more than 8000 cases of respiratory infections and 774 deaths in 30 countries on all continents until its containment in July 2003. Subsequently, MERS-CoV gave rise to another epidemic in September 2012, which started in Saudi Arabia and spread to 27 countries, causing more than 2500 cases of infections and 866 deaths [45,46,47].

Finally, in December 2019, a new coronavirus was identified, which was responsible for an outbreak of pneumonia in Wuhan, China. This virus (different from SARS-CoV and MERS-CoV) was named SARS-CoV-2 according to the International Committee on Taxonomy of Viruses (ICTV), and the WHO has named it coronavirus disease 2019 (COVID-19), the illness caused by this viral agent [48,49]. In the following months, SARS-CoV-2 transmission accelerated, and COVID-19 spread to several countries, with infections ranging from asymptomatic or mild upper respiratory tract illness to pneumonia and acute respiratory distress syndrome. The WHO officially declared the global outbreak of COVID-19 a pandemic on 11 March 2020 [50,51,52]. On 16 December 2022, the WHO reported over 647 million confirmed cases of COVID-19 and more than 6.6 million deaths worldwide [53].

## 3. Pharmacological Treatments for Respiratory Virus Infections

Viral respiratory infections represent one of the main causes of medical consultations in the world, so the prevention and treatment of these diseases remain an important objective. Antiviral therapy drugs prevent the activity of viral proteins involved in the multiple stages of the replication cycle of respiratory viruses, such as structural proteins or replication enzymes [54,55,56]. Antiviral drugs also play an important role in the prophylaxis and treatment of influenza infections. Ion channel protein (M2) blockers, NA inhibitors, and viral polymerase inhibitors are the three classes of antivirals approved for clinical use. The first drugs licensed for influenza treatment were adamantanes (amantadine and rimantadine), which act by blocking M2 ion channels. Nevertheless, adamantanes are only effective against influenza A viruses, and even resistant strains to these therapeutic agents have emerged, which are still in circulation. Therefore, adamantanes are no longer recommended in therapy against this disease [57,58].

Currently, the only class of antivirals appropriate for influenza therapy is NA inhibitors (oseltamivir, zanamivir, peramivir, and laninamivir). Although these drugs can inhibit all NA types and subtypes of influenza viruses by binding to the catalytic site of the enzyme, influenza A and B viruses resistant to NA inhibitors have also recently emerged [59,60]. In addition, viral polymerase inhibitors have emerged as novel compounds in influenza treatment. These drugs are baloxavir marboxil, favipiravir, and pimodivir. Some research has demonstrated that polymerase inhibitors have greater clinical efficacy compared to NA inhibitors. However, it has recently been shown that both baloxavir and pimodivir are associated with the emergence of new viral variants that are not very susceptible to these antivirals [29,61].

On the other hand, aerosolized ribavirin is the only antiviral drug licensed for the treatment of RSV infections, although its use is limited due to issues with efficacy, toxicity, and cost. Therefore, this drug has only been employed for the treatment of life-threatening infections in immunosuppressed patients. The use of ribavirin has not caused a significant decrease in mortality or duration of hospitalization in patients with RSV disease, so other alternatives for the treatment of RSV infections are being explored. Still, it will be several years until they are approved [62].

Unlike the previous viruses, there are currently no specific drugs approved for the prophylaxis and therapy of infections by HPIVs, HMPVs, HAdVs, HRVs, and HBoVs. In many cases, management of infections by these pathogens consists of supportive care to control symptoms or the off-label use of broad-spectrum antiviral agents, such as ganciclovir, cidofovir, or ribavirin for the treatment of HAdVs infections [63,64,65,66,67].

Likewise, there are no specific, effective, and clinically proven antiviral drugs against SARS-CoV and MERS-CoV, so therapy in patients infected with these viruses consists of supportive care, sometimes supplemented with different combinations of drugs. However, the efficacy of these treatments in SARS-CoV and MERS-CoV diseases remains unproven [68,69]. In this sense, although various research groups have made great efforts to develop an effective and safe treatment against COVID-19, a specific antiviral agent for the therapy of the disease caused by the novel SARS-CoV-2 has not yet been discovered [70]. Different studies have demonstrated that the drugs initially proposed for the treatment of COVID-19 are not effective against this disease, such as hydroxychloroquine, lopinavir/ritonavir, and ribavirin. Recently, Pfizer developed an antiviral drug called PF-07321332 (nirmatrelvir), which is an inhibitor of the 3C-like protease (3CL^pro^, also known as main protease or M^pro^) from SARS-CoV-2. The combination of nirmatrelvir and ritonavir (commercially known as PAXLOVID™) has been reported to reduce hospitalizations by 89% after administration within three days after symptom onset, according to a phase II/III clinical trial [71,72]. Nevertheless, the development of additional effective and specific antivirals against SARS-CoV-2 proteins is important to provide therapies that reduce the severity of COVID-19, decrease the spread of the viral agent, and combat future pandemic CoVs [73,74]. The chemical structures of the different drugs used in the treatment of respiratory viral infections are shown in Figure 1.

Thus, the relatively limited number of approved antiviral drugs, the emergence of viral strains with resistance mutations to these therapeutic agents, and the associated side effects have increased the need to develop new alternative antiviral drugs. Antiviral strategies involve the search for molecules that block the stages of the viral lytic cycle, such as attachment and entry to the host cell, replication, transcription, and translation. Natural products isolated from plants and other sources have emerged as an important resource of biologically active compounds with novel chemical structures, which have shown activity against several molecular targets of respiratory viruses [75,76,77]. Likewise, structural modification and chemical synthesis have emerged as additional strategies for the development of these natural products, as well as known and newly derived compounds [78,79]. Among the molecules with activity against respiratory viruses are alkaloids, flavonoids, glycosides, lignans, polyphenols, saponins, and terpenoids. Of particular note, various studies have reported that quinones also possess biological properties over respiratory viruses [80,81,82].

## 4. Quinones

Quinones are molecules comprised of a basic benzoquinone chromophore, which is an unsaturated cyclic structure with two carbonyl groups. These compounds can be classified into three main classes: benzoquinones, naphthoquinones, and anthraquinones. As previously described, the benzoquinone-type structure forms the basic unit of quinones. Therefore, benzoquinones serve as an important building block in quinone biosynthesis. In nature, benzoquinones are found in flowering plants, fungi, lichens, and insects [83].

On the other hand, naphthoquinones are molecules structurally related to naphthalene. Naphthoquinone structure consists of a benzene ring linked to conjugated cyclic diketone, with the two carbonyl groups arranged in the *para* orientation (1,4-naphthoquinones) and rarely in the *ortho* orientation (1,2-naphthoquinones) [84,85]. It has been reported that 1,4-naphthoquinones can act as electrophiles in Michael-type additions due to the presence of a highly electron-deficient double bond in their structure [86]. Considering that the tautomeric forms of 1,4-naphthoquinones (named 1,4-naphthohydroquinones or 1,4-dihydroxynaphthalenes) can act as nucleophiles and couple with their corresponding naphthoquinones, dimeric naphthoquinones can also be found in nature [87]. Naphthoquinones are biosynthesized by a wide variety of organisms, including plants and fungi. Biosynthetic pathways include the *o*-succinylbenzoate pathway, the phenylpropanoid/mevalonic acid (MVA) pathway, the homogentisate/MVA pathway, the acetate-malonate pathway, and the futalosine pathway. The 1,4-naphthalenoid ring of biosynthesized naphthoquinones is usually linked to one or more -CH_3_, -OH, and/or -OCH_3_ groups and, in some compounds, to a liposoluble side chain [88,89].

The third class of quinones is anthraquinones, which are compounds derived from anthracene. The structure of anthraquinones is comprised of a planar three-ring aromatic system, with two carbonyl groups in the central ring. Thus, the basic structure of these molecules is known as 9,10-anthracenedione or 9,10-dioxoanthracene [83,90]. Among the natural quinones, anthraquinones are the largest group, with about 700 reported compounds. Most of these molecules have been isolated from plants, lichens, and fungi, although they are also distributed in microorganisms, insects, and animals [91,92]. However, these organisms use different biosynthetic pathways for the production of anthraquinones. For example, plants employ the shikimate and acetate-malonate pathways for the synthesis of these compounds. In contrast, fungi use the acetate-malonate pathway primarily for the synthesis of polyketide precursors. In nature, these quinone-type compounds can be found in either free or glycosidic forms. Likewise, anthraquinones can have various substituents such as -CH_3_, -OCH_3_, -OH, -CHO, -COOH, -CH_2_OH, and/or more complex groups. Anthraquinone derivatives include hydroanthraquinones (compounds obtained by the reduction of double bonds in the benzene ring), anthrols (analogs with one -OH group in the central ring), anthrones (derivatives with one carbonyl group in the central ring), dimeric anthraquinones, and naphthodianthrones [93,94,95]. The chemical structures of the different types of quinones are represented in Figure 2.

Quinones have interesting chemical and pharmacological properties. These molecules can undergo one- and two-electron reductions, generating semiquinones and hydroquinones, respectively. Subsequently, these structures can be oxidized and form quinones again, generating a cyclic redox system with the production of cytotoxic reactive oxygen species (ROS). Thus, quinones participate in various biological processes, such as oxidative reactions and electron transport during metabolic pathways [96,97].

In addition, quinones are considered primary structures since they have shown a wide variety of biological activities. Considering this, the quinone-type structure has been useful for the design and synthesis of new compounds with potential pharmacological properties. Hence, as primary structures, quinones and their derivatives could serve as active and selective ligands for multiple biological targets. The wide range of quinone biological properties covers anti-inflammatory, immunomodulatory, cardioprotective, hepatoprotective, neuroprotective, antiseizure, anticancer, antibacterial, antifungal, antiparasitic, and antiviral activities. Several investigations have reported that quinones possess bioactivity over numerous viruses such as hepatitis B and C, dengue, herpes simplex, human immunodeficiency virus (HIV), coxsackievirus A16 (CVA16), and poliovirus. Finally, the activity of different types of quinones (both natural and synthetic) against respiratory viruses and their molecular viral targets has also been described, including various strains of influenza viruses and CoVs [81,82,98,99,100].

## 5. Quinones and Respiratory Viruses

Several studies have exhibited the inhibitory properties of the different types of quinones against a wide variety of respiratory viruses and their molecular targets. These investigations focused on the activity of naphthoquinones, anthraquinones, and other quinone-type derivatives against influenza A and B viruses. Likewise, quinones also showed inhibitory properties against CoVs (SARS-CoV and SARS-CoV-2) and respiratory viruses such as PIV, RSV, and HRV. The antiviral activities of all types of quinones are summarized in Table 1 and fully described in the following sections.

### 5.1. Quinones with Activity against Influenza Viruses

Various studies have reported the anti-influenza activity of naphthoquinone-type molecules isolated from different natural sources. The chemical structures of these natural quinones (compounds **1**–**19**) are shown in Figure 3.

The antiviral activity of rhinacanthins C (**1**), D (**2**), N (**3**), and Q (**4**) from the roots of *Rhinacanthus nasutus* (a medicinal plant belonging to the Acanthaceae family and employed for the treatment of herpes virus infections) was assessed in infected cells. All compounds inhibited the activity of the influenza virus A/PR/8/34 (H1N1), with mean inhibitory concentration (IC_50_) values of 0.30, 0.95, 1.95, and 23.7 µM for **1**, **2**, **3**, and **4**, respectively. Previously, the molecules did not show significant cytotoxicity in Vero cells, with mean cytotoxic concentration (CC_50_) values of 25.89 (**1**) and >50 µM (**2**, **3**, and **4**) [101].

Other works have investigated the potential of naphthoquinones in the inhibition of influenza virus molecular targets, such as the NA, a viral surface glycoprotein that facilitates the release of the newly synthesized virions from the host cell surface by removing sialic acids [142]. This enzyme is not specific to viruses and is also found in bacteria. Both bacterial and viral NAs recognize terminal sialic acid residues on the surface of host cells and cleave the α-2,3 glycosidic bonds of these residues. Therefore, some research groups use in vitro assays with NA from bacterial sources (for example, *Clostridium perfringens*) in the search for compounds with antiviral activity and employ tests with substrates that are recognized by viral and bacterial NAs. However, caution must be exercised when comparing the IC_50_ values obtained in inhibition assays with both sources [143,144].

The monomeric naphthoquinone 2-methoxy-6-acetyl-7-methyljuglone (**5**) obtained from the roots of *Polygonum cuspidatum* (a Chinese medicinal herb from the Polygonaceae family with various uses) evidenced an inhibitory activity on NA from *C. perfringens*, with IC_50_ of 8.9 µM [102]. In another research, the anti-NA activity of shikometabolins E (**6**) and F (**7**) was evaluated. The results showed that these dimeric naphthoquinones isolated from the roots of *Lithospermum erythrorhizon* (a perennial herb from the Boraginaceae family with red pigments that are used as dyestuffs in different products) inhibited the functions of the NA from *C. perfringens*, with IC_50_ values of 1.91 µg/mL for **6** and 2.79 µg/mL for **7** [103]. *L. erythrorhizon* roots also contain 1,4-naphthoquinones such as shikonin (**8**) and its derivatives acetylshikonin (**9**), isobutylshikonin (**10**), deoxyshikonin (**11**), β,β-dimethylacrylshikonin (**12**), and β-hydroxyisovalerylshikonin (**13**). These molecules were tested in two NA inhibition assays with sialidases from glycosyl hydrolase (GH) family 33 (*C. perfringens*) and GH34 (influenza virus A/Bervig_Mission/1/18 H1N1). All the natural products exhibited inhibitory activity over the bacterial NA, with IC_50_ of 53.8 (**8**), 2.5 (**9**), 2.9 (**10**), 27.5 (**11**), 1.9 (**12**), and 3.4 µM (**13**). Likewise, these naphthoquinones inhibited the activity of recombinant viral sialidase, with IC_50_ of 34.1 (**8**), 41.4 (**9**), 40.5 (**10**), 63.4 (**11**), 47.3 (**12**), and 40.5 µM (**13**) [104].

In addition to NA glycoprotein, several investigations have explored the potential effect of naphthoquinones on other viral targets, either in computational tests or in vitro experiments. One of these targets is the PA (polymerase acidic) subunit of viral RNA polymerase, which contains an endonuclease active pocket in its N-terminal domain that participates in viral transcription and replication [145]. Recent work revealed the inhibitory activity of some molecules over the N-terminal domain of the influenza A virus PA subunit (the protein was obtained from coding RNA DNA of viral strain A/California/07/09 H1N1) in a novel assay based on AlphaScreen technology (amplified luminescent proximity assay system). Among the evaluated compounds, an endonuclease inhibition was reported by lapachol (**14**), as well as mompain (**15**) and quambalarine B (**16**) obtained from the fungus *Quambalaria cyanescens* (IC_50_ values of 19, 0.43, and 0.29 µM, respectively). In the same study, an X-ray crystallography test evidenced that **16** binds to the N-terminal domain of the PA protein through its 7,8-dihydroxynaphthoquinone moiety and ketone moiety [105].

On the other hand, flexible docking and molecular dynamic simulations have demonstrated that juglone (**17**) was able to interact with active sites of the NA and HA (the most abundant glycoprotein on the viral surface; HA initiates infection by recognizing host cell-surface glycoconjugates with sialic acid as receptors and then by HA-mediated fusion of viral and host cellular membranes). The 2-cyclohexene-1,4-dione moiety of **17** bound to HA from the influenza A(H5N1) virus through hydrophobic interactions with residues Ile155, His183, and Tyr195. The 2-cyclohexene-1,4-dione ring was also bound to NA key binding site residues (Arg156 and Arg292) via electrostatic interactions. Strong H-bond interaction between the -OH group of **17** and the carboxylate group of Glu276 was observed [106,146,147]. According to in silico studies, plumbagin (**18**) can also bind specifically to the active sites of HA and NA proteins of influenza virus A/2009 (H1N1), as well as the M2 ion channel protein, which is involved in the virion entry and assembly of new infectious particles [107,148].

Likewise, our research group has contributed to the knowledge about the antiviral properties of naphthoquinones by isolating the zeylanone epoxide (**19**) and evaluating its activity against influenza A and B viruses. Secondary metabolite **19** was obtained from the stem bark of *Diospyros anisandra*, an endemic plant of the Yucatan Peninsula rich in quinone-type compounds, belonging to the Ebenaceae family and employed in traditional Mayan medicine for the treatment of skin problems [149]. Subsequently, in vitro assays showed that this dimeric naphthoquinone has no significant cytotoxicity on Madin-Darby canine kidney (MDCK) cells at concentrations ≤ 12.5 μM (CC_50_ of 21.70 µM) and has activity against four influenza viruses: A/Yucatan/2370/09 (H1N1)pdm (IC_50_ of 0.65 µM), A/Mexico/InDRE797/10 (H1N1-H275Y)pdm (IC_50_ of 2.77 µM), A/Sydney/5/97 (H3N2; IC_50_ of 1.6 µM), and B/Yucatan/286/10 (IC_50_ of 2.22 µM). Then, time-of-addition experiments were used to establish the replication cycle stage at which **19** acts. The results suggested that compound **19** could act either by direct binding to the virus or by blocking some stage after viral adsorption [108].

Finally, zeylanone epoxide **19** caused a decrease in the number of gene segment copies that encodes the viral nucleoprotein (NP). Therefore, the effect of **19** on the NP intracellular distribution was evaluated by indirect immunofluorescence assays (IFA) at 4, 6, and 8 h after infection with influenza A(H1N1) and A(H3N2) viruses. The results indicated that NP remained in the nucleus of the infected cells. However, the number of infected cells was lower in infection with influenza A(H1N1) viruses than with A(H3N2) viruses [108]. NPs are important viral proteins since they confer stability to the viral ribonucleoprotein particles (vRNPs) and participate in their transport. In addition, since NP is the major protein component of vRNPs, the distribution of this viral protein functions as a localization marker for vRNPs, since NP is found predominantly in the nucleus of infected cells in the early stages of infection. In the late stages of infection, NP accumulates in large amounts in the cytoplasm [150,151,152]. Considering this, the authors highlighted that **19** could inhibit the formation of daisy-chain complexes, interrupting the export of vRNPs to the cytoplasm. Nevertheless, the investigation concluded that the evaluation of other proteins involved in the transport of vRNPs and their interaction with **19** is necessary [108].

Several studies have focused on the synthesis of naphthoquinones and the evaluation of their activity against influenza viruses. The structures of the anti-influenza synthetic naphthoquinones **20**–**27** are represented in Figure 4.

Synthetic naphthoquinone derivatives **20** and **21** (2-substituted- and 2,3-disubstituted-1,4-naphthoquinones) were obtained from naphthazarin (**22**; a natural product present in plants of the families *Boraginaceae*, *Droseraceae*, and *Nepenthaceae*) and lawsone (**23**; a secondary metabolite found in leaves and flowers of *Lawsonia inermis*), respectively. Both synthetic compounds showed in vitro antiviral activity against a strain of influenza A virus (52% for **20** and 50% for **21**) [109,153]. Another molecule with anti-influenza activity is **24**, which was synthesized starting from **25** (a natural compound from *Tabebuia avellanedae*). The fluoride-derivative **24** inhibited the infection of MDCK cells by swine influenza virus A/Iowa/15/30 (H1N1), with 35% of inhibition [110].

In another research, 3,3′-(arylmethylene)bis(2-hydroxy-1,4-naphthoquinone) analogs (**26a–m**) were synthesized from **23** and substituted aromatic aldehydes. These dimeric derivatives were evaluated with in vitro assays against two NA (*C. perfringens* and influenza A H5N1 virus). All molecules inhibited bacterial NA (percentages of inhibition ranging from 70.9 to 96.6%). Then, the ten analogs with percentages of inhibition >80% were assessed in viral NA inhibition assay; molecules **26a** and **26b** exhibited the lowest IC_50_ values (29 and 26.5 µM, respectively). Further, the docking simulation evidenced that compound **26b** interacts with amino acids of binding pocket from the NA of influenza A(H5N1) virus through H-bonds with Arg118, Arg371, Tyr406, Glu277, Asp151, and Arg152, as well as hydrophobic interactions with Tyr347 [111].

Likewise, the synthesis and antiviral activity of (R)-1-(5,8-dihydroxy-1,4-dioxo-1,4-dihydronaphthalen-2-yl)-4-methylpent-3-en-1-yl-3-(1H-indol-3-yl) propanoate (**27**) has been reported. This esterified derivative of **8** promoted cell viability in A549 (human lung carcinoma) and MDCK cells infected with influenza virus A/PR/8/34 (H1N1), with CC_50_ values of 316 and 730 µg/mL, respectively. Compound **27** also reduced viral yield and inhibited influenza virus A/PR/8/34 (H1N1) replication in a dose-dependent manner. Then, the authors evaluated the effect of the synthesized naphthoquinone on other viral targets and found that **27** inhibited the viral NA activity. Molecular docking analysis showed that **27** could bind specifically to the active site of NA through H-bond interactions with the -NH groups of Arg118, Arg152, and Arg371 and the -OH group of Glu227. Finally, **27** caused a decrement in expression levels of viral NP mRNA in infected cells [112,154].

The anti-influenza activity of several anthraquinones has also been investigated. The structures of these molecules (anthraquinones **28**–**47**) are shown in Figure 5.

In a previous study, the antiviral activity of aloe-emodin (**28**) and aloe-emodin acetate (**29**) was evaluated in infected MDCK cells. These anthraquinones were isolated from the leaves of *Cassia roxburghii* (a medicinal plant from the Fabaceae/Leguminosae family employed due to its laxative and purgative properties), and the results evidenced an inhibitory activity against influenza virus A/WSN/33 (H1N1), with IC_50_ values of 2.00 (**28**) and 10.23 µg/mL (**29**), as well as CC_50_ values of 0.47 (**28**) and 1.32 µg/mL (**29**). The authors concluded that the antiviral effect of **28** and **29** could be attributed to the number of -OH groups in these structures [113].

Moreover, the antiviral activity of anthraquinone **28** and two derivatives was assessed against another influenza virus strain (A/Taiwan/CMUH01/07 H1N1). After evaluation on MDCK cells, compounds exhibited CC_50_ values of 76.6, 25.7, and 18.3 μg/mL for **28**, emodin (**30**), and chrysophanol (**31**), respectively. Although the three metabolites were demonstrated to reduce the cytopathic effect (CPE) in infected MDCK cells, compound **28** showed the strongest inhibition of virus yield, with an IC_50_ value of less than 0.05 μg/mL. Proteomic analysis and Western blot (WB) indicated that **28** up-regulating galectin-3 in MDCK cells, which is involved in the induction of interferon γ (IFN-γ) and β (IFN- β). Quantitative PCR and WB confirmed that this anthraquinone also up-regulates the galectin-3 expression [114].

In another investigation, anthraquinone **30** was isolated from the roots of *Polygonatum odoratum* (a herbaceous plant belonging to the Liliaceae family and used to treat diabetes or rheumatic heart disease), along with physcion (**32**) and a new derivative called polygodoquinone A (**33**; a naphthoquinone analog linked to an anthraquinone via a C-C bond). The three anthraquinones showed inhibitory activity against influenza virus A/WSN/33 (H1N1), with IC_50_ values of 11.0, 11.4, and 2.3 µM for **30**, **33**, and **32**, respectively. The compounds exhibited CC_50_ values of 36.5 (**33**), 79.5 (**30**), and 94.0 µM (**32**) after cytotoxicity evaluation on 293 T-Gluc cells [115].

Furthermore, the cytotoxic and anti-influenza activities of **28**, **30**, **32**, emodin-1-O-β-D-glucopyranoside (**34**), chrysophanol 8-O-glucoside (**35**), rhein 8-glucoside (**36**), and aloe-emodin-8-O-β-D-glucopyranoside (**37**) have been determined. All compounds had no significant cytotoxicity in A549 and MDCK cells (<25 μg/mL). Likewise, these anthraquinones inhibited the activity of influenza virus A/PR/8/34 (H1N1) at concentrations ranging from 12.5 to 25 μg/mL. Natural product **30** also demonstrated to inhibit the activity of several human and avian influenza A virus strains at 6.25–25 μg/mL, which were A/PR/8/34 (H1N1), A/ShanTou/16/09 (H1N1), A/ShanTou/1233/06 (H1N1), A/ShanTou/602/06 (H3N2), A/ShanTou/364/05 (H3N2), A/Quail/HongKong/G1/97 (H9N2), A/Chicken/Guangdong/A1/03 (H9N2), and A/Chicken/Guangdong/1/05 (H5N1). Finally, the authors concluded that the pharmacological mechanism of **30** could be attributed to the regulation of various markers involved in the PPARα/γ-AMPK pathway and fatty acid metabolism after numerous biological tests [116].

Another work revealed that **30** had no cytotoxic effects on MDCK cells (IC_50_ of 182.95 μg/mL) and inhibited the activity of influenza virus A/ShanTou/169/06 (H1N1), with a mean maximal effective concentration (EC_50_) of 4.25 μg/mL. This natural product significantly reduced influenza A virus-induced up-regulation of mRNA and protein expressions of MyD88, TRAF6, and Toll-like receptors (TLRs) such as TLR2, TLR3, TLR4, and TLR7. Metabolite **30** caused a decrement in phosphorylations of p38, MAPK/JNK, and translocation of nuclear factor κB (NF-κB) and regulated several markers involved in oxidative stress. Likewise, **30** inhibited viral replication, lung edema, and inflammatory response in vivo. Finally, anthraquinone **30** enhanced the activation of the Nrf2 signaling pathway, which can inhibit oxidative stress and suppress the activation of the aforementioned signaling pathways during influenza infection [117]. In this sense, there is evidence that anthraquinones inhibited pro-inflammatory markers such as cytokines and nitric oxide (NO) in ovalbumin-induced asthma mouse models. Anthraquinones reduced the infiltration of inflammatory cells (macrophages and eosinophils) and pulmonary tissue injuries in these in vivo models, too. Thus, the effects of anthraquinones on the immune system may be related to other complications associated with viral respiratory infections, such as acute asthma [155].

A new hydroanthraquinone (6-O-demethyl-4-dehydroxyaltersolanol A, **38**) and four other known derivatives were isolated from the culture broth of the fungal strain *Nigrospora* sp. YE3033, previously obtained from *Aconitum carmichaeli* root. A CPE inhibition assay demonstrated that compound **38**, 4-dehydroxyaltersolanol A (**39**), and altersolanol B (**40**) inhibited the activity of influenza virus A/PR/8/34 (H1N1), with IC_50_ values of 2.59, 8.35, and 7.82 µg/mL, respectively. Moreover, cytotoxicity on MDCK cells was determined, and CC_50_ values of 94.92 (**38**), 90.71 (**39**), and 33.62 µg/mL (**40**) were obtained [118].

Another anthraquinone with anti-influenza activity is rhein (**41**), which is found in traditional medicinal plants such as *Rheum palmatum*, *Aloe barbadensis*, *Cassia angustifolia*, and *Polygonum multiflorum*. This molecule showed a CC_50_ of 64.59 µg/mL in A549 cells and inhibited the proliferation of influenza virus A/ShanTou/169/06 (H1N1) according to a plaque inhibition assay (EC_50_ of 1.51 µg/mL). Time-of-addition test suggested that **41** could inhibit influenza virus adsorption and replication. Other biological experiments evidenced that **41** could suppress influenza virus-induced oxidative stress in vitro by reduction, increment, or up-regulation of various markers. Anthraquinone **41** decreased the expressions of several TLRs, phosphorylations of p38, Akt, MAPK/JNK, and translocation of NF-κB. In addition, **41** inhibited in vitro expression of pro-inflammatory cytokines and matrix metalloproteinases, as well as pulmonary inflammation and histopathological changes in vivo. Thus, the authors concluded that **41** could inhibit viral replication by suppressing oxidative stress and different signaling pathways induced by the influenza virus [119].

Previously, our research group isolated aloesaponarin-I (**42**) and aloesaponarin-II (**43**) from the roots of *Aloe vera* (a plant with diverse therapeutic properties) and synthesized analogs by methylation, acetylation, and O-glycosyl reactions starting from **42**. Although compounds had no significant cytotoxicity in MDCK cells (CC_50_ > 100 µM), only derivatives **44** and **45** showed a reduction in CPE against two influenza virus strains: A/Yucatan/2370/09 (H1N1), with IC_50_ of 30.77 for **44** and 13.70 μM for **45**; A/Mexico/InDRE797/10 (H1N1), with IC_50_ of 62.28 for **44** and 19.47 μM for **45**. Then, the authors carried out time-of-addition assays to assess the effect of these derivatives during one cycle of replication (0–10 h). Both anthraquinones caused a decrement in viral yields when added 6–10 h post-infection, so this study concluded that the tetra-O-acetyl-β-D-glucopyranosyl substituent at the C3 position of **44** and **45** might affect the activity of the influenza A(H1N1) viruses [120].

Finally, some anthraquinones have been investigated for their anti-NA activity, such as **30**, **32**, and **34**. These metabolites and anthraquinone glycosides physcion-8-O-β-D-glucopyranoside (**46**) and emodin-8-O-β-D-glucopyranoside (**47**) were obtained from the roots of *P. cuspidatum*. Anthraquinone glucosides **34**, **46**, and **47** showed much better inhibition of NA from *C. perfringes* than their aglycones. Compounds **30** and **32** had IC_50_ values of 5.4 and >200 μM, while **34**, **46**, and **47** had IC_50_ values of 0.43, 6.2, and 0.85 μM, respectively [102].

Several investigations have also focused on evaluating the anti-influenza activity of other types of quinones, such as anthrones, hydroquinones, and benzoquinones. The structures of these quinone derivatives (compounds **48**–**55**) are shown in Figure 6.

In this sense, hypericin (**48**; an aromatic polycyclic anthrone present in *Hypericum triquetrifoliurn*) exhibited a virucidal effect against influenza virus A/Brazil at concentrations ranging from 3.12 to 50 µg/mL [121]. Compound **48** and its analogs dibromohypericin (**49**), tetrabromohypericin (**50**), and gymnochrome B (**51**) were evaluated against the influenza A virus. All molecules showed antiviral activity, with minimum 100% inhibitory concentrations (MIC_100_) of 13 (**48**), <5 (**49**), 250 (**50**), and 78 nM (**51**) [122].

Other quinone derivatives with antiviral activity are hydroquinones, particularly 1,4-hydroquinone (**52**). This compound was isolated from the leaves of *Elaeocarpus tonkinensis* (a medicinal plant from Vietnam that belongs to the Elaeocarpaceae family) and inhibited the activity of influenza viruses A/PR/8/34 (H1N1), A/HongKong/8/68 (H3N2), and B/Lee/40, with EC_50_ values of 31.9, 19.7, and 54.3 µg/mL, respectively. Previously, **52** did not show significant cytotoxicity on MDCK cells (CC_50_ > 300 µg/mL) [123].

In another study, *tert*-butylhydroquinone (**53**) was shown to form a complex with H14 from influenza virus A/mallard/Astrakhan/263/82. According to X-ray crystallography analyses, compound **53** was bound to the protein at the interface between two monomers that make up the HA trimer. Considering that HA is composed of three monomers (each monomer consists of HA1 and HA2 subunits bound by a disulfide bond), this viral protein has three binding sites for **53**. Quinone **53** was bound to HA mainly through hydrophobic interactions with interface residues, which consist of residues from the long HA2 α-helices of each monomer (Leu291, Leu982, Ala1012, Leu552, and Leu992) [124,146].

The cytotoxicity and anti-influenza activity of **53** and some derivatives have also been tested. Hydroquinone **53** and *tert*-butylbenzoquinone (**54**) exhibited antiviral effects against a pseudovirus expressing H7 HA protein in its viral envelope, although the antiviral activity was higher for **53** (IC_50_ of 6 μM) than for **54** (IC_50_ > 50 μM), suggesting that -OH groups play an important role in the antiviral properties of **53**. Cytotoxicity assay with 293T cells evidenced a CC_50_ value >270 μM for **53**. Molecular dynamic simulations demonstrated that **53** was able to interact with active sites of H7 via H-bonds between the -OH groups with Arg54 and Glu97. The *tert*-butyl group of **53** was also bound to H7 through hydrophobic interactions with residues Leu55, Leu98, and Leu99 [125].

On the other hand, embelin (**55**) is a benzoquinone that has shown activity against the influenza virus A/PR/8/34 (H1N1). This natural product was isolated from the fruits of *Embelia ribes* (a plant from the Myrsinaceae family and used to treat diabetic ulcers or bronchitis), and its cytotoxicity in MDCK cells and antiviral activity were assessed (CC_50_ of 3.1 µM and IC_50_ of 0.3 µM). Metabolite **55** was evaluated against five other influenza A and B viruses, obtaining IC_50_ values of 0.1 (A/mallard/Pennsylvania/10218/84 H5N2), 0.5 (A/California/07/09 H1N1pdm), 0.5 (A/Vladivostok/02/09 H1N1), 0.6 (A/Aichi/2/68 H3N2), and 0.2 µM (B/Malaysia/2506/04). Time-of-addition experiments were carried out, and results showed that **55** was most effective when added at the early stages of the viral replication cycle (0–1 h post-infection). Hemagglutination inhibition assay demonstrated that **55** prevent agglutination of erythrocytes induced by influenza virus A/PR/8/34 (H1N1; hemagglutination titer was 1:32), so in silico docking simulations were performed. Compound **55** was bound to avian H5 HA through H-bond between the -OH group with Tyr91. Likewise, the -OH and carbonyl groups of **55** interacted with Arg193, Ser227, Gly228, and Glu190 from human H5 HA protein through H-bonds, while a π-alkyl and π-anion interactions between Leu194, Glu190, and **55** were observed [126].

### 5.2. Quinones with Activity against CoVs

Unlike influenza viruses, studies with quinones and their activity against CoVs are limited. Some quinones have exhibited inhibitory properties against SARS-CoV and its protein targets. The chemical structures of these compounds (quinones **56**–**67**) are represented in Figure 7.

One of the interesting molecular targets of SARS-CoV is 3CL^pro^, which has a conserved structure among CoVs (despite sequence variation). 3CL^pro^ and another protease (papain-like cysteine protease or PL^pro^) cleave two polyproteins (pp1a and pp1ab) and produce various nonstructural proteins implicated in viral genome transcription and replication [156]. The activity of aloe-emodin (**28**) against 3CL^pro^ has been previously reported. Anthraquinone **28** exerted inhibitory effects on the 3CL^pro^ cleavage activity from SARS-CoV by cell-free and cell-based cleavage assays, with IC_50_ values of 132 and 366 µM, respectively. This compound did not show significant cytotoxicity in Vero cells, with CC_50_ of 11,592 µM [127].

Spike (S) protein has been another attractive viral target of CoVs since it is involved in host cell entry. This glycoprotein consists of two subunits (the S1 subunit, which binds to the host cell receptor angiotensin-converting enzyme 2 or ACE2, and the S2 subunit, which leads the fusion of the viral and host cell membranes) and requires to be cleaved by the transmembrane protease/serine subfamily member 2 (TMPRSS2) to trigger its functions [157]. Anthraquinones have also shown inhibitory effects on this protein. In this sense, emodin (**30**) inhibited the interaction of the S protein with ACE2 in a dose-dependent manner (IC_50_ of 200 µM), while rhein (**41**) slightly inhibited the interaction between the protein and the receptor. Then, the inhibitory effect of **30** on the interaction of SARS-CoV S protein with Vero E6 cell receptors was determined using an S protein-pseudotyped retrovirus. Compound **30** blocked the interaction between protein S and Vero E6 cells, as well as reduced infectivity of S protein-pseudotyped retrovirus in a dose-dependent manner [128].

Another work showed that molecule **30** was able to inhibit the functions of the SARS-CoV and HCoV-OC43 3a ion channels (a cation-selective channel comprised of the 3a protein that plays an important role in SARS-CoV release from the infected cells) expressed in *Xenopus* oocytes. Likewise, **30** inhibited HCoV-OC43 release from infected Rhabdomyosarcoma (RD) cells [129].

Other quinone-type derivatives have been demonstrated to inhibit the activity of the above-mentioned viral targets of SARS-CoV. For example, tanshinone I (**56**), tanshinone IIA (**57**), tanshinone IIB (**58**), methyl tanshinonate (**59**), cryptotanshinone (**60**), dihydrotanshinone I (**61**), and rosmariquinone (**62**) exerted an inhibitory effect over 3CL^pro^ in a dose-dependent manner (except for **60**), with IC_50_ values of 38.7, 89.1, 24.8, 21.1, 226.7, 14.4, and 21.1 µM, respectively. These tanshinones (diterpene quinolones) were previously isolated from the roots of *Salvia miltiorrhiza* (an Asian medicinal plant belonging to the Laminaceae family and employed to treat coronary heart disease) and also inhibited the activity of PL^pro^ from SARS-CoV in a time-dependent manner. These natural products evidenced potent inhibition after 60 min of preincubation (IC_50_ of 8.8 for **56**, 1.6 for **57**, 10.7 for **58**, 9.2 for **59**, 0.8 for **60**, 4.9 for **61**, and 30.0 µM for **62**) [130].

Celastrol (**63**), pristimerin (**64**), tingenone (**65**), iguesterin (**66**), and dihydrocelastrol (**67**) are quinone methide triterpenes with activity against 3CL^pro^. Compounds **63**, **64**, **65**, and **66** were obtained from the bark of *Triterygium regelii* (a plant from the Celastraceae family used in traditional Chinese medicine to treat inflammatory and autoimmune diseases), while **67** was synthesized from **63**. All quinone methide triterpenes exhibited inhibitory activity over 3CL^pro^, with IC_50_ values of 10.3 (**63**), 5.5 (**64**), 9.9 (**65**), 2.6 (**66**), and 21.7 µM (**67**). In the same investigation, molecular docking simulation showed that the -OH group of **66** bound to the active site of 3CL^pro^ from SARS-CoV via H-bonds with the carbonyl group of Cys44 and -OH group of Thr25 [131].

Due to the global health emergency derived from the current pandemic, various studies have focused on the search for compounds with activity against the novel SARS-CoV-2 and its molecular targets. Among the molecules evaluated are quinones, and the chemical structures of these compounds (quinones **68**–**83**) are shown in Figure 8.

Molecular docking analyses have demonstrated that different types of quinones are able to interact with 3CL^pro^ residues from SARS-CoV-2. The -OH and carbonyl groups of embelin (**55**) bound to Leu141, Gly143, Ser144, His163, and Glu166 through H-bonds, while π-sulfur, alkyl, and π-alkyl interactions between Cys145 and His41 residues with **55** were observed. On the other hand, anthraquinone **30** was bound to 3CL^pro^ through H-bond between its carbonyl group with Glu166, as well as Asn142 through its two hydrogen atoms. Vitamin K1 (**68**; phylloquinone form of vitamin K) and coenzyme Q10 (**69**) also interacted with Gly143 and Asn142 through H-bonds, while H-bond interactions were noted between the carbonyl group of the quinone methides methylprednisolone (**70**) and dexamethasone (**71**) and the -NH group of Gly143 [132].

In another study, molecular mechanics and dynamics calculations evidenced that -OH and carbonyl groups of quinone methide **63** were able to interact with 3CL^pro^ active site via H-bonds with Thr25 and His41. Furthermore, Cys145, Met49, and Pro168 residues were bound to **63** through covalent and π bonds [133]. In silico simulations showed that *ortho*-quinone derivatives **72**, **73**, and **74** of clovamide (a caffeoyl conjugate found in *Theobroma cacao*) bound to 3CL^pro^ residues. The -OH and carbonyl groups of **72** interacted with Gly143, Ser144, Cys145, Glu166, and Val186 through H-bonds, while a π-sulfur interaction between Cys145 and the quinone ring was observed. Similar to **72**, analog **73** was bound to 3CL^pro^ through H-bonds between Asn142, Gly143, His163, Glu166, and Thr190 with -OH and carbonyl groups. Pro168 and Gln189 bound to the quinone ring of **73** by π interactions. Finally, the -OH and carbonyl groups of derivative **74** interacted with amino acids His41, Cys145, His163, and Phe140 through H-bonds; Cys145 and Pro168 were able to bind to quinone rings through π-alkyl interactions [134].

Docking simulations have also shown that terrequinone A (**75**), zeylanone (**76**), and carminic acid (**77**) interact with the active site of 3CL^pro^ from SARS-CoV-2. Quinone **75** was bound to Gly143, Glu166, Arg188, and Gln189 through H-bonds. Compound **75** was docked to 3CL^pro^ through van der Waals (Thr26, Asn142, and His163), π-alkyl (His41, Cys145, Glu166, and Gln189), and hydrophobic (Met49 and Met165) interactions. Dimeric naphthoquinone **76** was able to interact with His163, His164, and Glu166 through H-bonds, as well as Gly143, Cys145, and Gln189 by van der Waals-type interactions. In the same work, anthraquinone **77** was demonstrated to bind 3CL^pro^ by H-bond (Glu166), van der Waals (Thr24, Thr26, Asn142, and Gly143) and hydrophobic (Met49) interactions, while quinone methide triterpene **66** interacted with Glu166 from the protease through H-bond [135].

In addition to in silico studies, in vitro assays have shown that quinones possess activity against SARS-CoV-2 and 3CL^pro^. The 3CL^pro^ inhibitory effects of synthetic naphthoquinones and anthraquinones were determined, and compounds 7-methyl juglone ethyl acetate (**78**), 5-(benzyloxy)-7-methyl-1,4-naphthoquinone (**79**), propionyl juglone (**80**), 1,4-naphthoquinone (**81**), and 2-acetyl-8-methoxy-1,4-naphthoquinone (**82**) exhibited potent inhibition, with IC_50_ values of 220.9, 160.68, 129.77, 110.13, and 72.07 nM, respectively. Later, the authors performed molecular docking analyses with juglone (**17**), **80**, and the most potent 3CL^pro^ inhibitor, **82**. The phenolic -OH and the carbonyl group of **17** interacted with Gly143 and Glu166 from 3CL^pro^ through H-bonds, while **80** was bound to the protease by H-bond interactions between the -NH group of Gly143 and the -OH group of Ser144 with the carbonyl group of **80**. Naphthoquinone **82** was bound to the imidazole moiety of His41, -NH group of Gly143, and -NH group of Glu166 through H-bonds with the carbonyl, acetyl, and methoxy groups, respectively. Cytotoxicity of naphthoquinones was also tested over human normal fibroblast cells (HFF-1), and the least cytotoxic molecules were lawsone (**23**), vitamin K3 (**83**), **79**, **81**, and **82** (IC_50_ values of >50, >50, 32.9, 26.7, and 41.2 µM). Moreover, **82** did not show significant cytotoxicity on Vero E6 cells at concentrations <20 µM (cell viability >90%). Considering this, the activity of **82** against SARS-CoV-2 was assessed in infected cells, obtaining an EC_50_ of 4.55 µM [136].

Finally, the antiviral activity of five herbal formulations containing anthraquinones **28**, **30**, **41**, chrysophanol (**31**), physcion (**32**), and other compounds was evaluated against pseudo-typed SARS-CoV-2 in infected HEK293T cells. Three formulations inhibited the activity of the pseudo-typed viral particles, with percentages of infectivity ranging from 0.00 to 0.03% [137].

### 5.3. Quinones with Activity against Other Respiratory Viruses

In addition to influenza viruses and CoVs, there are some studies that reported the antiviral effect of quinones on other respiratory viruses. The structures of these molecules (quinones **84**–**88**) are represented in Figure 9.

Previously, the activity of emodin (**30**) and the anthraquinone derivatives hypericin (**48**), emodin anthrone (**84**), and emodin bianthrone (**85**) were evaluated against PIV type-3. The concentrations required to reduce the virus titer (1 log_10_) were 0.1 (**48**), >10 (**30** and **84**), and 4 µg/mL (**85**) [138]. Compound **30** has been demonstrated to inhibit RSV activity. This anthraquinone from *R. palmatum* had no significant cytotoxicity on human laryngeal carcinoma cells (HEp-2) at low concentrations (CC_50_ of 76.783 µmol/L). The natural product **30** reduced CPE in infected HEp-2 cells (>80% of inhibition in replication of RSV at 30 µmol/L). Furthermore, when **30** was added post-infection, this metabolite inhibited RSV activity according to the results of MTT and plaque reduction assays (EC_50_ values of 14.27 and 13.06 µmol/L, respectively). Time-of-addition experiments demonstrated that **30** inhibited the replication of RSV when added 0–4 h post-inoculation so that this compound could affect the early stages of the viral replication cycle. Quantitative PCR also showed that **30** increased mRNA levels of IFN-γ and decreased tumor necrosis factor α (TNF-α) mRNA expression in infected HEp-2 cells [139]. Vitamin E quinone (**86**) obtained from the stems of *Celastrus hindsii* also decreased the CPE in infected HEp-2 cells with RSV A2 strain (IC_50_ of 3.13 µM) [140].

Finally, the anti-HRV activity of several quinones has been reported. The aforementioned rhinacanthins C (**1**), D (**2**), N (**3**), and Q (**4**) from the roots of *R. nasutus* caused an inhibition on the HRV-1B activity, with IC_50_ values of 0.29, 0.24, 0.97, and 5.35 µM, respectively [101]. Later, the effect of quinones against 3C^pro^ from HRV was demonstrated since the quinone analogs **87** and **88** inhibited the activity of recombinant 3C^pro^ (IC_50_ of 0.85 for **87** and 8.4 µM for **88**), while SDS-PAGE analysis verified that **87** and **88** completely suppressed the catalytic activity of the protease (5 µM for **87** and 50 µM for **88**). Then, flexible docking simulations were carried out and showed that these derivatives interacted with the active site of 3C^pro^ from HRV-14 [141].

## 6. Future Trends

In the present review, the activity of a large number of quinones against various respiratory viruses and their molecular targets has been described. Among the evaluated pathogens were influenza viruses and CoVs. On the other hand, investigations with PIVs, RSVs, and HRVs were very limited, while no studies were found showing that quinones inhibit the activity of HMPVs, AdVs, and BoVs. Some quinones demonstrated activity against more than one virus. Then, considering that these compounds possess a wide variety of chemical structures, it is likely that quinones or their derivatives also inhibit the replication of less common respiratory viruses for which there are no reports.

As previously described, various types of quinones have been evidenced to be active against a wide range of influenza viruses and CoVs. Several investigations have shown that quinone-type molecules inhibit protein targets of influenza viruses (NA, HA, M2, PA, and NP) and CoVs (3CL^pro^, PL^pro^, and S protein). However, it is necessary to broaden the spectrum of viral targets, determining whether quinones inhibit the synthesis or activity of other proteins involved in the replication cycle of influenza viruses (PB1, PB2, NEP, NS1, and M1), CoVs (membrane protein, nucleocapsid protein, and other accessory molecules), and other respiratory viruses, either through in silico models or in vitro tests. In this sense, there are many computational tools and molecular modeling studies that could be useful to assess the interaction mode of compounds with their therapeutic target. Likewise, it is important to establish whether the evaluated quinone participates in other cellular processes, which could allow the cell to protect itself from viral infection or secrete molecules that eliminate the infectious agent.

In summary, in silico analyses and in vitro assays would help to establish the mechanism of action of the evaluated quinones, identifying at which stage of the viral replication cycle they act. Later, research needs to be carried out using in vivo tests and clinical trials with pharmacokinetics, toxicity, and efficacy studies. Structure–activity relationship studies can also be performed to find the molecule with optimal activity. Moreover, it is very important to continue the search for quinones with potential activity against respiratory viruses (whether in plants, marine organisms, or other natural sources), characterize their structures completely, determine their antiviral activity with the models already described and, as a consequence, develop quinones with improved activity through chemical synthesis or other methods.

## 7. Conclusions

For many years, respiratory viruses have represented a global public health problem, resulting in annual economic losses and numerous pandemics, including the current pandemic caused by SARS-CoV-2. This situation has motivated researchers to search for new and promising antiviral molecules from natural sources or synthetics. Although a large number of quinones with potent antiviral activity have been reported, none of them are currently used as drugs to treat viral respiratory diseases. There is still research to be done in the field of quinones with activity against respiratory viruses. Therefore, it is important to continue the work at different levels (in silico, in vitro, and in vivo) until reaching the clinical trials, taking advantage of the versatile quinone scaffold for the development of future drugs against respiratory viruses.

## Figures and Tables

**Figure 1 molecules-28-01981-f001:**
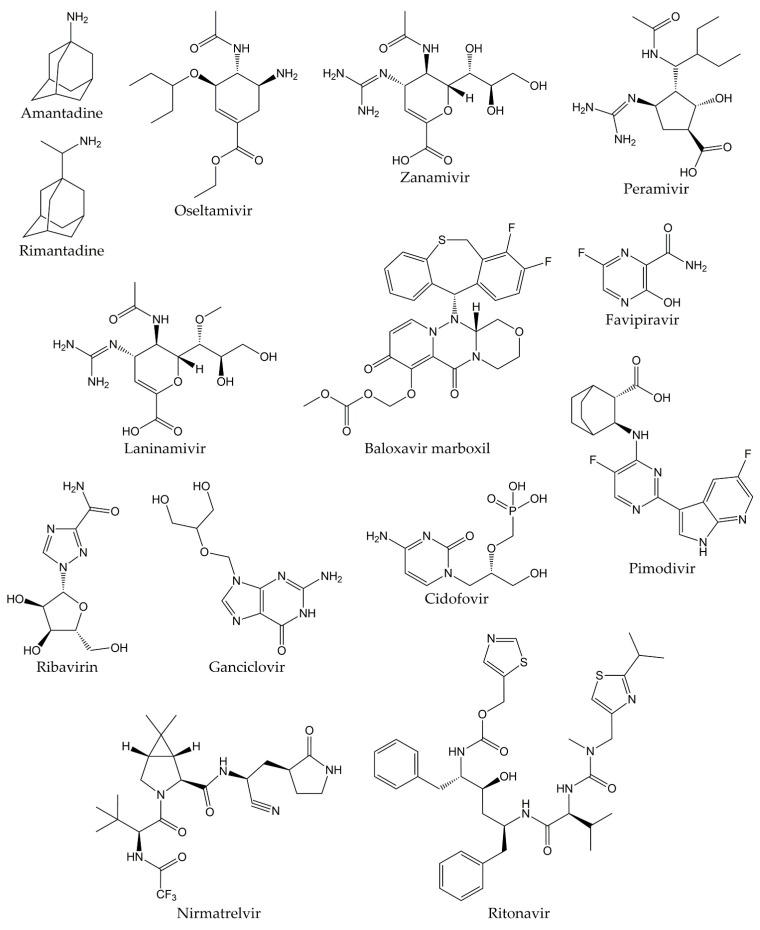
Chemical structures of antivirals against respiratory viruses.

**Figure 2 molecules-28-01981-f002:**
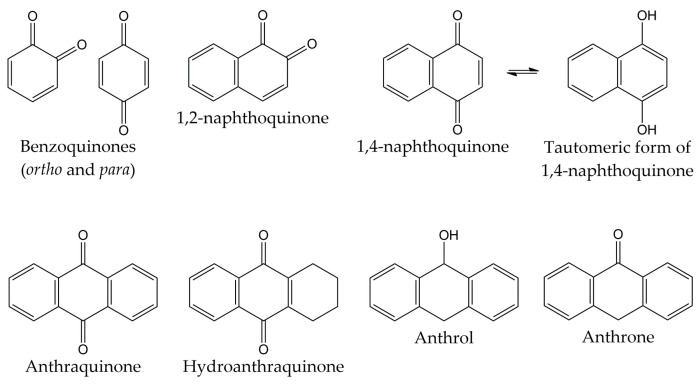
Chemical structures of the different types of quinones.

**Figure 3 molecules-28-01981-f003:**
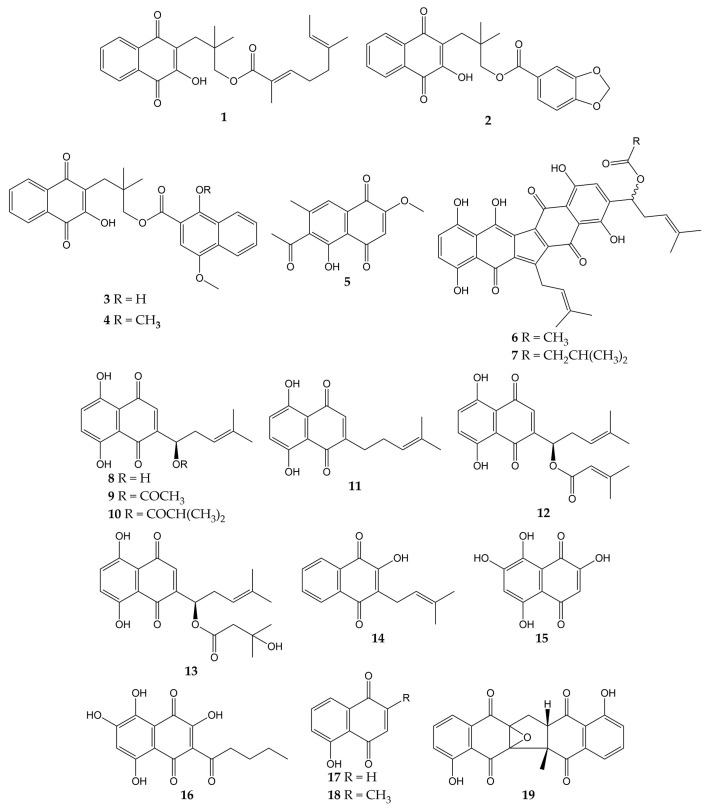
Chemical structures of natural naphthoquinones **1**–**19**.

**Figure 4 molecules-28-01981-f004:**
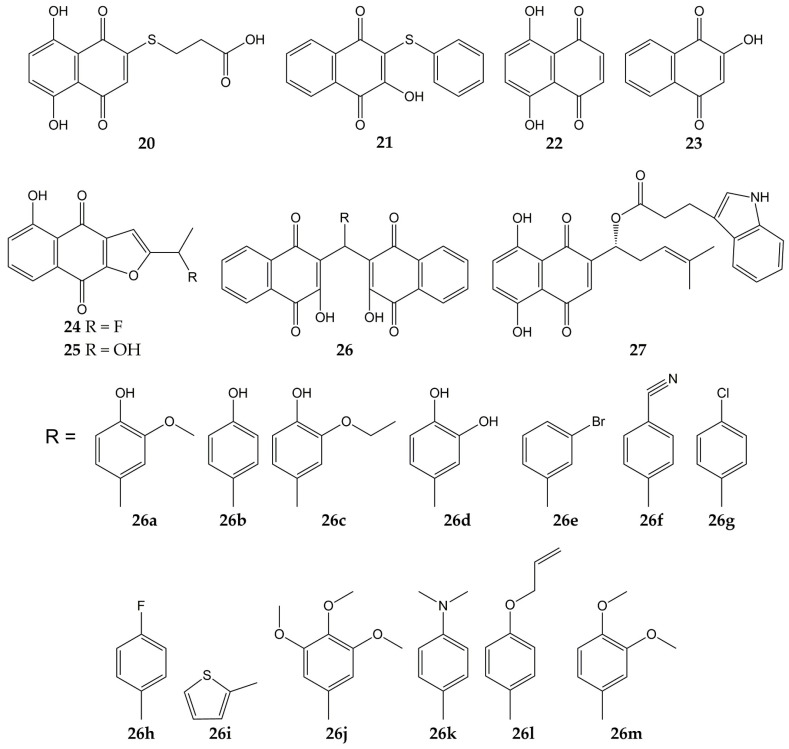
Chemical structures of synthetic naphthoquinones **20**–**27**.

**Figure 5 molecules-28-01981-f005:**
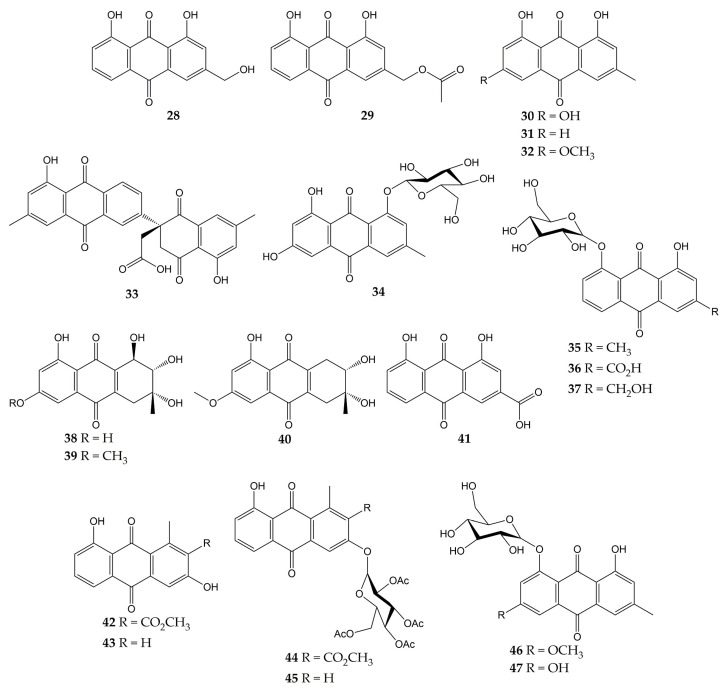
Chemical structures of natural and synthetic anthraquinones **28**–**47**.

**Figure 6 molecules-28-01981-f006:**
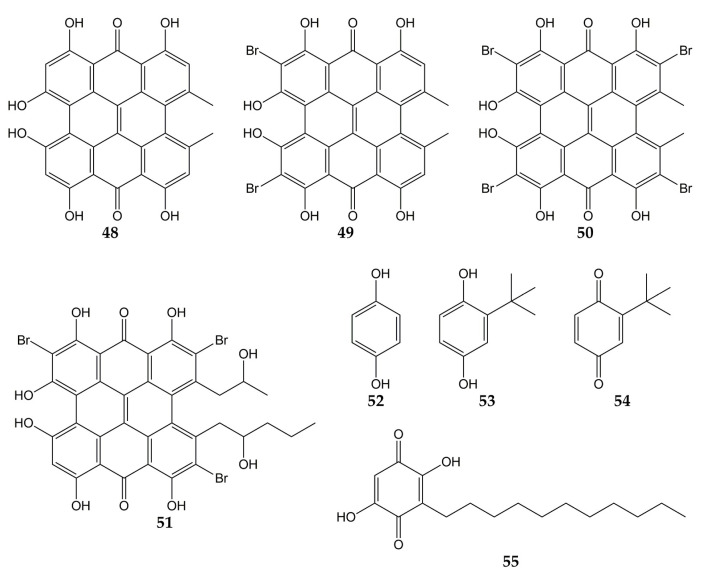
Chemical structures of quinones **48**–**55**.

**Figure 7 molecules-28-01981-f007:**
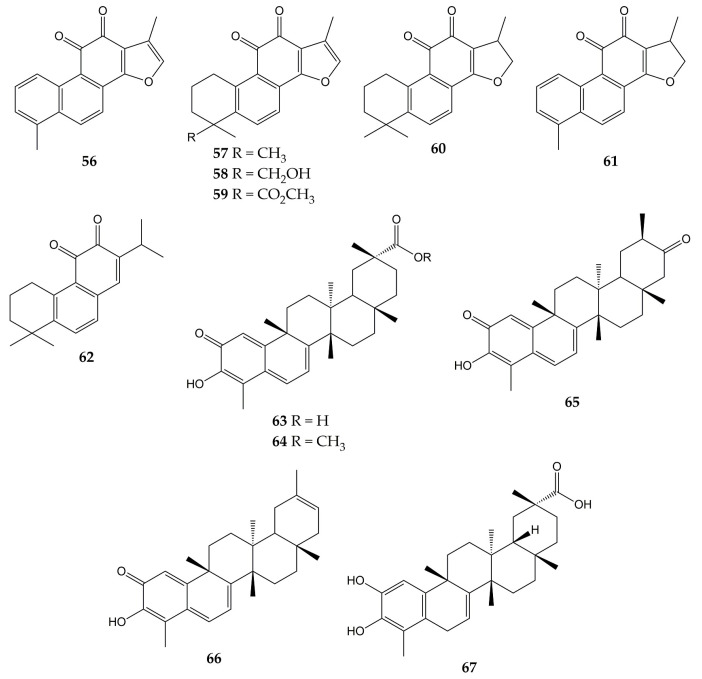
Chemical structures of quinones **56**–**67**.

**Figure 8 molecules-28-01981-f008:**
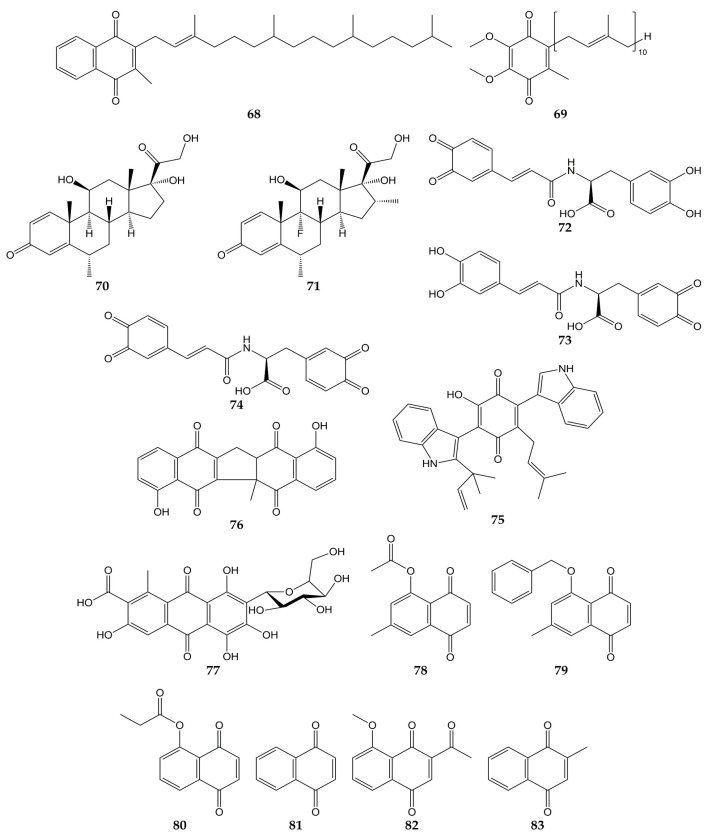
Chemical structures of quinones **68**–**83**.

**Figure 9 molecules-28-01981-f009:**
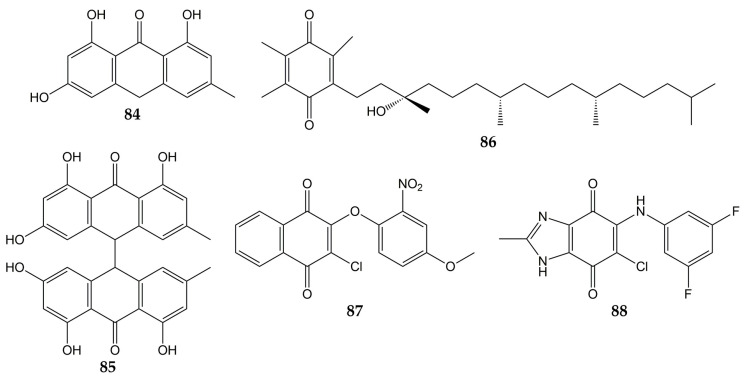
Chemical structures of quinones **84**–**88**.

**Table 1 molecules-28-01981-t001:** Quinones with activity against respiratory viruses and their protein viral targets.

Compound(s)	Source	Virus/Target	Effect	Ref.
Rhinacanthins C (**1**), D (**2**), N (**3**), and Q (**4**)	*Rhinacanthus nasutus* roots	A/PR/8/34 (H1N1) HRV-1B	IC_50_ = 0.30–23.7 µM IC_50_ = 0.24–5.35 µM	[101]
2-methoxy-6-acetyl-7-methyljuglone (**5**), emodin (**30**), physcion (**32**), emodin-1-O-β-D-glucopyranoside (**34**), physcion-8-O-β-D-glucopyranoside (**46**), emodin-8-O-β-D-glucopyranoside (**47**)	*Polygonum cuspidatum* roots	NA (*Clostridium perfringens*)	IC_50_ = 0.43->200 μM	[102]
Shikometabolins E (**6**) and F (**7**)	*Lithospermum erythrorhizon* roots	NA (*C. perfringens*)	IC_50_ = 1.91 and 2.79 µg/mL (**6** and **7**)	[103]
Shikonin (**8**), acetylshikonin (**9**), isobutylshikonin (**10**), deoxyshikonin (**11**), β,β-dimethylacrylshikonin (**12**), β-hydroxyisovalerylshikonin (**13**)	*L. erythrorhizon* roots	NA (*C. perfringens*) NA (A/Bervig_Mission/1/18 H1N1)	IC_50_ = 1.9–63.4 µM	[104]
Lapachol (**14**), mompain (**15**), quambalarine B (**16**)	*Quambalaria cyanescens* fungus (**15** and **16**) and commercial compound (**14**)	PA (A/California/07/09 H1N1)	IC_50_ = 0.29–19 µM Interaction of **16** with PA	[105]
Juglone (**17**)	Not given	HA and NA (H5N1)	Interaction with proteins	[106]
Plumbagin (**18**)	Not given	HA, NA, and M2 (A/2009 H1N1)	Interaction with proteins	[107]
Zeylanone epoxide (**19**)	*Diospyros anisandra* stem bark	A/Yucatan/2370/09 (H1N1)pdm A/Mexico/InDRE797/10 (H1N1-H275Y)pdm A/Sydney/5/97 (H3N2) B/Yucatan/286/10 NP (H1N1 and H3N2)	IC_50_ = 0.65–2.77 µM Decrease of RNA and inhibition of nuclear NP export	[108]
Substituted naphthoquinones (**20** and **21**)	Synthesis	Influenza A virus	Inhibition of 52% and 50% (**20** and **21**)	[109]
Fluoride derivative (**24**)	Synthesis	A/IOWA/15/30 (H1N1)	Inhibition of 35%	[110]
Dimeric derivatives (**26a**–**m**)	Synthesis	NA (*C. perfringens*) NA (H5N1)	Inhibition of 70.9–96.6% IC_50_ = 29 and 26.5 µM (**26a** and **26b**) Interaction of **26b** with NA	[111]
Esterified derivative of **8** (**27**)	Synthesis	Infected A549 and MDCK cells A/PR/8/34 (H1N1) NA and NP (H1N1)	CC_50_ = 316 and 730 µg/mL Reduction of viral yield Interaction of **27** with NA Decrease of NP mRNA levels	[112]
Aloe-emodin (**28**), emodin acetate (**29**)	*Cassia roxburghii* leaves	A/WSN/33 (H1N1)	IC_50_ = 2.00 and 10.23 µg/mL (**28** and **29**)	[113]
Compounds **28**, **30**, chrysophanol (**31**)	Not given	A/Taiwan/CMUH01/07 (H1N1)	Reduction of CPE Up-regulation of galectin-3	[114]
Compounds **30**, **32**, polygodoquinone A (**33**)	*Polygonatum odoratum* roots	A/WSN/33 (H1N1)	IC_50_ = 2.3–11.4 µM	[115]
Compounds **28**, **30**, **32**, **34**, chrysophanol 8-O-glucoside (**35**), rhein 8-glucoside (**36**), aloe-emodin-8-O-β-D-glucopyranoside (**37**)	Commercial compounds	A/PR/8/34 (H1N1) A/ShanTou/16/09 (H1N1) A/ShanTou/1233/06 (H1N1) A/ShanTou/602/06 (H3N2) A/ShanTou/364/05 (H3N2) A/Quail/HongKong/G1/97 (H9N2) A/Chicken/Guangdong/A1/03 (H9N2) A/Chicken/Guangdong/1/05 (H5N1).	Inhibition of A/PR/8/34 (H1N1) activity at 12.5–25 μg/mL (all compounds) Inhibition of influenza A viruses activities at 6.25–25 μg/mL (**30**) Regulation of several markers of the PPARα/γ-AMPK pathway and fatty acid metabolism (**30**)	[116]
Compound **30**	Commercial compound	A/ShanTou/169/06 (H1N1)	EC_50_ = 4.25 μg/mL Regulation of several markers involved in oxidative stress, inflammation, and different signaling pathways during influenza infection	[117]
6-O-demethyl-4-dehydroxyaltersolanol A (**38**), 4-dehydroxyaltersolanol A (**39**), altersolanol B (**40**)	*Nigrospora* sp. YE3033 from *Aconitum carmichaeli* root	A/PR/8/34 (H1N1)	IC_50_ = 2.59–8.35 µg/mL	[118]
Rhein (**41**)	Commercial compound	A/ShanTou/169/06 (H1N1)	EC_50_ = 1.51 µg/mL Regulation of several markers involved in oxidative stress, inflammation, and different signaling pathways during influenza infection	[119]
Derivatives of aloesaponarin-I **42** (**44** and **45**)	Synthesis	A/Yucatan/2370/09 (H1N1) A/Mexico/InDRE797/10 (H1N1)	IC_50_ = 13.70–62.28 μM Decrease in viral yields	[120]
Hypericin (**48**)	Commercial compound	A/Brazil	Virucidal effect at 3.12–50 µg/mL	[121]
Compound **48**, dibromohypericin (**49**), tetrabromohypericin (**50**), gymnochrome B (**51**)	Commercial (**48**), synthetic (**49** and **50**), and natural (**51**) compounds	Influenza A virus strain	MIC_100_ = <5–250 nM	[122]
1,4-hydroquinone (**52**)	*Elaeocarpus tonkinensis* leaves	A/PR/8/34 (H1N1) A/HongKong/8/68 (H3N2) B/Lee/40	EC_50_ = 19.7–54.3 µg/mL	[123]
*tert*-butylhydroquinone (**53**)	Not given	H14 HA (A/mallard/Astrakhan/263/82)	Interaction with HA	[124]
Compound **53**, *tert*-butylbenzoquinone (**54**)	Commercial compounds	Pseudovirus expressing H7 HA	IC_50_ = 6 and >50 μM (**53** and **54**) Interaction of **53** with HA	[125]
Embelin (**55**)	*Embelia ribes* fruits	A/PR/8/34 (H1N1) A/California/07/09 (H1N1)pdm A/Vladivostok/02/09 (H1N1) A/Aichi/2/68 (H3N2) A/mallard/Pennsylvania/10218/84 (H5N2) B/Malaysia/2506/04 HA (H5N2)	IC_50_ = 0.1–0.6 µM Hemagglutination inhibitory effect (titer 1:32) Interaction with HA	[126]
Compound **28**	Commercial compound	3CL^pro^ (SARS-CoV)	IC_50_ = 132 and 366 µM	[127]
Compounds **30** and **41**	Commercial compounds	S protein (SARS-CoV) Pseudovirus expressing S protein	IC_50_ = 200 µM Inhibition of interaction Reduction of infectivity	[128]
Compound **30**	Commercial compound	3a protein (SARS-CoV and HCoV-OC43)	Inhibition of viruses release	[129]
Tanshinone I (**56**), tanshinone IIA (**57**), tanshinone IIB (**58**), methyl tanshinonate (**59**), cryptotanshinone (**60**), dihydrotanshinone I (**61**), rosmariquinone (**62**)	*Salvia miltiorrhiza* roots	3CL^pro^ (SARS-CoV) PL^pro^ (SARS-CoV)	IC_50_ = 0.8–226.7 µM	[130]
Celastrol (**63**), pristimerin (**64**), tingenone (**65**), iguesterin (**66**), dihydrocelastrol (**67**)	*Triterygium regelii* bark (**63**, **64**, **65**, and **66**) and synthetic compound (**67**)	3CL^pro^ (SARS-CoV)	IC_50_ = 2.6–21.7 µM Interaction with 3CL^pro^	[131]
Compounds **30**, **55**, vitamin K1 (**68**), coenzyme Q10 (**69**), methylprednisolone (**70**), dexamethasone (**71**)	Not given	3CL^pro^ (SARS-CoV-2)	Interaction with 3CL^pro^	[132]
Compound **63**	Commercial compound	3CL^pro^ (SARS-CoV-2)	Interaction with 3CL^pro^	[133]
Clovamide derivatives (**72**, **73**, **74**)	Not given	3CL^pro^ (SARS-CoV-2)	Interaction with 3CL^pro^	[134]
Terrequinone A (**75**), zeylanone (**76**), carminic acid (**77**)	Not given	3CL^pro^ (SARS-CoV-2)	Interaction with 3CL^pro^	[135]
Compound **17**, 7-methyl juglone ethyl acetate (**78**), 5-(benzyloxy)-7-methyl-1,4-naphthoquinone (**79**), propionyl juglone (**80**), 1,4-naphthoquinone (**81**), 2-acetyl-8-methoxy-1,4-naphthoquinone (**82**)	Synthetic compounds	3CL^pro^ (SARS-CoV-2) SARS-CoV-2	IC_50_ = 72.07–220.9 nM (**78**–**82**) Interaction with 3CL^pro^ (**17**, **80**, **82**) Inhibition of SARS-CoV-2 (**82**; EC_50_ = 4.55 µM)	[136]
Compounds **28**, **30**, **31**, **32**, **41**	Compounds present in herbal formulations	Pseudo-typed SARS-CoV-2	Inhibition of infectivity	[137]
Compounds **30**, **48**, emodin anthrone (**84**), emodin bianthrone (**85**)	Not given	PIV type-3	Reduction of viral titer	[138]
Compound **30**	*Rheum palmatum*	RSV	Reduction of CPE Inhibition of RSV activity	[139]
Vitamin E quinone (**86**)	*Celastrus hindsii* stems	RSV A2	IC_50_ = 3.13 µM	[140]
Derivatives **87** and **88**	Compounds reported in the Korea Chemical Bank	Recombinant 3C^pro^ (HRV) 3C^pro^ from HRV-14	IC_50_ = 0.85 and 8.4 µM (**87** and **88**) Interaction with 3C^pro^ from HRV-14	[141]

IC_50_: Mean inhibitory concentration. CC_50_: Mean cytotoxic concentration. EC_50_: Mean maximal effective concentration. MIC_100_: Minimum 100% inhibitory concentration. NA: Neuraminidase. HA: Hemagglutinin. M2: Ion channel protein. PA: Polymerase acidic. NP: Nucleoprotein. CPE: Cytopathic effect. S protein: Spike protein. 3CL^pro^: 3C-like protease. PL^pro^: Papain-like cysteine protease.

## Data Availability

Not applicable.

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
