# Peer review of "Quinones as Promising Compounds against Respiratory Viruses: A Review"

_molecules, 2023, doi:10.3390/molecules28041981_

Round 1

Reviewer 1 Report

GENERAL COMMENTS:

The manuscript presents an extensive review of the role of quinones against respiratory viruses which in my opinion is a valid contribution for researchers working in the search of compounds against these viruses. However I think that there some aspects need to be clearer in the Introduction, especially from section 1 to the end of section 2.1., which is the weakest part of this work.  The manuscript would also gain much in clarity and readability by carrying out some changes which I mention below (Table and figure mentions in the text, see below). There are also smaller errors that should be corrected and which will markedly improve the work.

CORRECTIONS TO BE CARRIED OUT:

Abstract: I would say that “Respiratory viruses represent a world public health problem…”, not that they “…have represented a world public health problem”, since they problem continues and will continue in the future, as far as we can predict.

Introduction, lines 41-42: please correct the sentence “Even, viruses cause 95% and 40% of all respiratory infections in children and older adults, respectively”, to “Viruses cause 95% and 40% of all respiratory infections in children and older adults, respectively”. The “even” is probably a wrong translation and is wrong in this context, in the beginning of a sentence.

Lines 42-47: the authors must specify the timeframe and geography of these data – what do this average of deaths and other data refer to? For example, these are data from the 20th century, or between 1950-2000, in the U.S. or in Europe, all over the world?  And 13-31% of deaths caused by viruses seems too vague; 13% from year x to year y, and 31% for another range of years, or maybe 13% in one location and 31% in another?

Line 67: the authors start another sentence with an “Even” that is grammatically wrong in the beginning of a sentence. Removing the “even” will result in a correct sentence.

Lines 73-74: as previously mentioned, the authors must specify the context – what “different parts of the world” reported 42,464 cases? And it is hard to believe that between 1995 and 2016 only 41 deaths due to these viruses were reported (where?). And if only the virus only caused 41 deaths between 1995-2016, it doesn´t seem to be worth mentioning.

Line 86: remove “on the other hand”, which would doesn’t make sense in the beginning of a section or subsection.

Line 179: the authors mean: “Antiviral drugs also paly an important role…”, right?

Lines 187 and following sentences: When the authors refer the current therapies, they should mention what chemical groups these molecules belong to (and even better, represent the structures); this is useful to compare with the quinones the manuscript is proposing to treat these viral infections

Line 231: remove “Then,” from the beginning of the sentence.

In table 1, in column “Source”, what does “---" mean? Unknown? Not specified in the reference? In this table some compounds were extracted from natural sources, others are commercial; the meaning of “---” should be explained in a footnote to Table 1.

REORGANIZATION  TO IMPROVE CLARITY:

I suggest the following reorganization to improve clarity. In section “5. Quinones and respiratory viruses”, it is difficult to fully understand the immense amount of information without restructuring it in the following way: when the authors describe the effect of each compound against different types of virus, it would be much clearer if they started by referring Table 1, which summarizes what is being described, and this Table should be in the first part of this section, which then describes the compounds and respective effects. The authors should also call attention to the figures with the chemical structures of the compounds in the beginning of each section and not at the end (e.g., “The chemical structures of 1-19 are shown in Figure 2 “ should be in the beginning of the text where they describe the activity of natural naphthoquinones and not in line 390, and the same for all the other types of quinone (Figs.3, 4, 5, 5 and 7, respectively).

Author Response

Response to Reviewer 1 Comments

Point 1: Abstract: I would say that “Respiratory viruses represent a world public health problem…”, not that they “…have represented a world public health problem”, since they problem continues and will continue in the future, as far as we can predict.

Response 1: We made the correction (line 10).

Point 2: Introduction, lines 41-42: please correct the sentence “Even, viruses cause 95% and 40% of all respiratory infections in children and older adults, respectively”, to “Viruses cause 95% and 40% of all respiratory infections in children and older adults, respectively”. The “even” is probably a wrong translation and is wrong in this context, in the beginning of a sentence.

Response 2: We corrected the sentence (line 41).

Point 3: Lines 42-47: the authors must specify the timeframe and geography of these data – what do this average of deaths and other data refer to? For example, these are data from the 20th century, or between 1950-2000, in the U.S. or in Europe, all over the world?  And 13-31% of deaths caused by viruses seems too vague; 13% from year x to year y, and 31% for another range of years, or maybe 13% in one location and 31% in another?

Response 3: We made the correction and updated the data, specifying that they are data from the WHO (2019) and that they apply all over the world (lines 43-48).

Point 4: Line 67: the authors start another sentence with an “Even” that is grammatically wrong in the beginning of a sentence. Removing the “even” will result in a correct sentence.

Response 4: We corrected the sentence (line 76).

Point 5: Lines 73-74: as previously mentioned, the authors must specify the context – what “different parts of the world” reported 42,464 cases? And it is hard to believe that between 1995 and 2016 only 41 deaths due to these viruses were reported (where?). And if only the virus only caused 41 deaths between 1995-2016, it doesn´t seem to be worth mentioning.

Response 5: We made the suggested corrections and specified the countries. Besides, we removed the number of deaths (lines 83-85).

Point 6: Line 86: remove “on the other hand”, which would doesn’t make sense in the beginning of a section or subsection.

Response 6: We corrected the sentence (line 97).

Point 7: Line 179: the authors mean: “Antiviral drugs also play an important role…”, right?

Response 7: We corrected the sentence (line 190).

Point 8: Lines 187 and following sentences: When the authors refer the current therapies, they should mention what chemical groups these molecules belong to (and even better, represent the structures); this is useful to compare with the quinones the manuscript is proposing to treat these viral infections.

Response 8: According to current drugs, we added the chemical structures and the groups to which these compounds belong. However, some molecules have various functional groups and do not belong to any category (such as NA inhibitors or polymerase inhibitors), unlike amantadine and rimantadine, which are tricyclic compounds belonging to the adamantanes (lines 198-240). We also included a figure (Figure 1, line 240).

Point 9: Line 231: remove “Then,” from the beginning of the sentence.

Response 9: We corrected the sentence (line 245).

Point 10: In table 1, in column “Source”, what does “---" mean? Unknown? Not specified in the reference? In this table some compounds were extracted from natural sources, others are commercial; the meaning of “---” should be explained in a footnote to Table 1.

Response 10: We removed “---” and added: “not given”. For better comprehension, we added a footnote with the main abbreviations of the table (lines 325-328).

Point 11: REORGANIZATION TO IMPROVE CLARITY: I suggest the following reorganization to improve clarity. In section “5. Quinones and respiratory viruses”, it is difficult to fully understand the immense amount of information without restructuring it in the following way: when the authors describe the effect of each compound against different types of virus, it would be much clearer if they started by referring Table 1, which summarizes what is being described, and this Table should be in the first part of this section, which then describes the compounds and respective effects. The authors should also call attention to the figures with the chemical structures of the compounds in the beginning of each section and not at the end (e.g., “The chemical structures of 1-19 are shown in Figure 2 “should be in the beginning of the text where they describe the activity of natural naphthoquinones and not in line 390, and the same for all the other types of quinone (Figs.3, 4, 5, 5 and 7, respectively).

Response 11: We followed the recommendations to improve the clarity of the text, which consisted of moving Table 1 at the beginning of section “5. Quinones and respiratory viruses” (line 324) and adding an introductory text in this section (lines 317-323). Besides, we moved Figures 3, 4, 5, 6, 7, 8, and 9 at the beginning of their respective section (lines 332, 430, 464, 567, 621, 676, and 741, respectively).

We appreciate all your comments and suggestions.

Sincerely,  

Dra. Rocío de Lourdes Borges Argáez

Unidad de Biotecnología

CICY

Reviewer 2 Report

The manuscript reports a good level of data. A series of in silico and in vitro experiments were described to be conducted for detecting natural and synthetic quinones as promising compounds against respiratory viruses. Authors do not abuse the use of self-citations (only 1-2 are used).

However, there are some comments that are listed below.

The minor check is required for the English language and style:

Lines 15-17 (Abstract). It would be good to paraphrase these two sentences to better understand their essence:  'On the other hand, there are no specific drugs licensed for the treatment of other viral respiratory diseases. In this sense, natural products and their derivatives have appeared as promising alternatives in the search for new compounds with antiviral activity'

For instance: 'On the other hand, no specific drugs are licensed to treat other viral respiratory diseases. In this sense, natural products and their derivatives have appeared promising alternatives in searching for new compounds with antiviral activity. Besides their chemical properties, quinones have demonstrated interesting biological activities'

Line 24 – the term 'benzoquinone' could be also added to the list of keywords as well as 'natural compounds' and 'synthetic quinones'.

Line 50 – The aim of the study should be clearly formulated 'constituents on the prevention of breast cancer'

Lines 317, 318why is the bacterial microorganism mentioned if the manuscript is devoted to the antiviral properties of quinones? 'evidenced an inhibitory activity on bacterial NA (BNA) from Clostridium perfringens'

Why weren't there enough references published in the last 3 years (especially for 2022)? For example, in Table 1, only 1 source for 2022 is used ([148]). In general, the list of used sources for 2022 and 2023 in the text is less than 10%, so it should be supplemented a little. Such publications can be easily found in PubMed/Scopus databases. These additions using sources for 2021-2023 years would significantly strengthen the novelty of the chosen topic, for instance:

Xin, D., Li, H., Zhou, S., Zhong, H., & Pu, W. (2022). Effects of Anthraquinones on Immune Responses and Inflammatory Diseases. Molecules27(12), 3831. https://doi.org/10.3390/molecules27123831

Xu, Z., Huang, M., Xia, Y., Peng, P., Zhang, Y., Zheng, S., Wang, X., Xue, C., & Cao, Y. (2021). Emodin from Aloe Inhibits Porcine Reproductive and Respiratory Syndrome Virus via Toll-Like Receptor 3 Activation. Viruses13(7), 1243. https://doi.org/10.3390/v13071243

Sharad, S., & Kapur, S. (2021). Indian Herb-Derived Phytoconstituent-Based Antiviral, Antimicrobial and Antifungal Formulation: An Oral Rinse Candidate for Oral Hygiene and the Potential Prevention of COVID-19 Outbreaks. Pathogens (Basel, Switzerland)10(9), 1130. https://doi.org/10.3390/pathogens10091130

Author Response

Response to Reviewer 2 Comments

Point 1: Lines 15-17 (Abstract). It would be good to paraphrase these two sentences to better understand their essence:  'On the other hand, there are no specific drugs licensed for the treatment of other viral respiratory diseases. In this sense, natural products and their derivatives have appeared as promising alternatives in the search for new compounds with antiviral activity'.

For instance: 'On the other hand, no specific drugs are licensed to treat other viral respiratory diseases. In this sense, natural products and their derivatives have appeared promising alternatives in searching for new compounds with antiviral activity. Besides their chemical properties, quinones have demonstrated interesting biological activities'.

Response 1: We corrected both sentences (lines 15-18).

Point 2: Line 24 – the term 'benzoquinone' could be also added to the list of keywords as well as 'natural compounds' and 'synthetic quinones'.

Response 2: We added the keywords “benzoquinone”, “natural compounds” and “synthetic quinones” (lines 24, 25) and removed “antiviral activity” and “respiratory infections” since we can only use three to ten keywords.

Point 3: Line 50 – The aim of the study should be clearly formulated 'constituents on the prevention of breast cancer'.

Response 3: We added the aim of the review manuscript at the end of the introduction (lines 56-59).

Point 4: Lines 317, 318 – why is the bacterial microorganism mentioned if the manuscript is devoted to the antiviral properties of quinones? 'evidenced an inhibitory activity on bacterial NA (BNA) from Clostridium perfringens'.

Response 4: In the manuscript, we included works with quinones and their inhibitory effects on bacterial neuraminidase (NA) from Clostridium perfringens since some research groups employed this bacterial NA in the search for antiviral compounds. For better comprehension, we included a paragraph respecting this (lines 345-351).

Point 5: Why weren't there enough references published in the last 3 years (especially for 2022)? For example, in Table 1, only 1 source for 2022 is used ([148]). In general, the list of used sources for 2022 and 2023 in the text is less than 10%, so it should be supplemented a little. Such publications can be easily found in PubMed/Scopus databases. These additions using sources for 2021-2023 years would significantly strengthen the novelty of the chosen topic, for instance:

Xin, D., Li, H., Zhou, S., Zhong, H., & Pu, W. (2022). Effects of Anthraquinones on Immune Responses and Inflammatory Diseases. Molecules, 27(12), 3831. https://doi.org/10.3390/molecules27123831

Xu, Z., Huang, M., Xia, Y., Peng, P., Zhang, Y., Zheng, S., Wang, X., Xue, C., & Cao, Y. (2021). Emodin from Aloe Inhibits Porcine Reproductive and Respiratory Syndrome Virus via Toll-Like Receptor 3 Activation. Viruses, 13(7), 1243. https://doi.org/10.3390/v13071243

Sharad, S., & Kapur, S. (2021). Indian Herb-Derived Phytoconstituent-Based Antiviral, Antimicrobial and Antifungal Formulation: An Oral Rinse Candidate for Oral Hygiene and the Potential Prevention of COVID-19 Outbreaks. Pathogens (Basel, Switzerland), 10(9), 1130. https://doi.org/10.3390/pathogens10091130

Response 5: We appreciate your contributions to the role of quinones and their activity against respiratory viruses. Unfortunately, the works that we report are the most recent and relevant that we find in the different databases (Google Scholar, Pubmed, and others). For this reason, in the manuscript, we highlight the importance of continuing to work with these molecules, since they have shown interesting properties over respiratory viruses. In our research group, we also continue to explore the antiviral activity of natural and synthetic quinones, in order to contribute more to this field of knowledge. We incorporated the suggested articles related to the immune system and SARS-CoV-2 as the manuscript focuses on human respiratory viruses or probable pandemic viruses, such as avian influenza viruses. The authors greatly appreciate your valuable input.

We appreciate all your comments and suggestions.

Sincerely,  

Dra. Rocío de Lourdes Borges Argáez

Unidad de Biotecnología

CICY

Reviewer 3 Report

This is a comprehensive review of the action of quinones against the various known human respiratory viruses.  The introduction to the respiratory viruses seems to cover the virtually all the human respiratory viruses with good references.  The authors then describe the presently available treatments, if any,  for the several classes of viruses and this also seems to be complete.  Then follows an extensive discussion of the structures of the various classes of quinones (benzoquinones, naphthoquinones and anthraquinones) with useful and accurate representations of the quinone structures with some of their most relevant substructures and descriptions of the effect of these agents on the respiratory viruses including infectivity and replication and possible toxicities on some mammalian cells. Specific chemical/biochemical mechanisms of action of these agents are not discussed (for instance, redox cycling and generation of reactive oxygen species)discussed in any detail.  Binding of some of the quinones to various amino acids of relevant proteins is discussed.  The effects of the quinones and the sources of the naturally-occurring quinones are heavily referenced.  Of particular helpful information is Table 1 which is a listing of the effects of quinones on the respiratory viruses, the origin of the quinones, the proper names of the quinones and the IC50 for the compounds on the specified viruses.  This table alone justifies the publication.  A more extensive discussion of the mechanisms of the biochemical reactions in relation to the structure and chemical properties of the quinones  would be helpful but should not be required for publication. I have listed below some very slight changes to the wording of the manuscript which might be helpful to the authors.

line 13 substitute "cause" for "originate"

line 19 substutute "including activity against" for "over"

line 23 "their" for 'its"

line 41 "Thus" for "Even"

line 45 "due to" for "for"

line 67 "However" for "Even"

line 128 "was responsible for" for "originated"

line 191 "In addition" for "Moreover"

line 231 "Also" for "Thus"

line 238 "Of particular note" for "However"

line 286 read "generating a cyclic redox system with the production of  cytotoxic reactive oxygen species (ROS).

line 289 "primary" for "privileged"

line 292 "primary" for privileged"

line 400 "against" for "over"

line 400 ...."most well-known"...

line 773 "are" for "was"

line 782 "On the other hand" for "Even"

line 782-783 omit "to have"

line 785 read "...the replication of less common respiratory viruses for which there are..."

line 808 "as a consequence" for "probably"

Author Response

Response to Reviewer 3 Comments

Point 1: line 13 substitute "cause" for "originate".

Response 1: We made the correction (line 13).

Point 2: line 19 substitute "including activity against" for "over".

Response 2: We made the correction (line 19).

Point 3: line 23 "their" for 'its".

Response 3: We made the correction (line 23).

Point 4: line 41 "Thus" for "Even".

Response 4: We made the correction following the suggestion of reviewer 1 (lines 41-48).

Point 5: line 45 "due to" for "for".

Response 5: Similar to point 4, we made the correction following the suggestion of reviewer 1 (lines 41-48).

Point 6: line 67 "However" for "Even".

Response 6: Similar to point 4, we made the correction following the suggestion of reviewer 1 (line 76).

Point 7: line 128 "was responsible for" for "originated".

Response 7: We made the correction (line 139).

Point 8: line 191 "In addition" for "Moreover".

Response 8: We made the correction (line 202).

Point 9: line 231 "Also" for "Thus".

Response 9: We made the correction (line 245).

Point 10: line 238 "Of particular note" for "However".

Response 10: We made the correction (line 253).

Point 11: line 286 read "generating a cyclic redox system with the production of cytotoxic reactive oxygen species (ROS).

Response 11: We made the correction (lines 300, 301).

Point 12: line 289 "primary" for "privileged".

Response 12: We made the correction (line 303).

Point 13: line 292 "primary" for privileged".

Response 13: We made the correction (line 306).

Point 14: line 400 "against" for "over".

Response 14: We made the correction (line 437).

Point 15: line 400 ...."most well-known"....

Response 15: We removed the sentence, so we omitted the correction (line 467).

Point 16: line 773 "are" for "was".

Response 16: We moved Table 1, so we made the correction in line 322.

Point 17: line 782 "On the other hand" for "Even".

Response 17: Similar to point 4, we made the correction following the suggestion of reviewer 1 (line 776).

Point 18: line 782-783 omit "to have".

Response 18: We made the correction (line 776).

Point 19: line 785 read "...the replication of less common respiratory viruses for which there are..."

Response 19: We made the correction (line 779).

Point 20: line 808 "as a consequence" for "probably".

Response 20: We made the correction (line 802).

We appreciate all your comments and suggestions.

Sincerely,  

Dra. Rocío de Lourdes Borges Argáez

Unidad de Biotecnología

CICY